

# AlignSAE: Concept-Aligned Sparse Autoencoders

**Minglai Yang**[†][‡]                                             *mingly@arizona.edu*
*University of Arizona*

**Xinyu Guo**                                                     *xinyuguo@arizona.edu*
*University of Arizona*

**Zhengliang Shi**                                               *zhenglis@andrew.cmu.edu*
*Carnegie Mellon University*

**Jinhe Bi**                                                     *bijinhe@outlook.com*
*LMU Munich*

**Steven Bethard**                                               *bethard@arizona.edu*
*University of Arizona*

**Mihai Surdeanu**[*]                                            *msurdeanu@arizona.edu*
*University of Arizona*

**Liangming Pan**[*]                                             *liangmingpan@pku.edu.cn*
*Peking University*

**Reviewed on OpenReview:** https://openreview.net/forum?id=I9UjKxW4nq

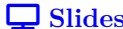 **Code & Data**    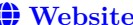 **Slides**    ⊕ **Website**

## Abstract

Large Language Models (LLMs) encode factual knowledge within hidden parametric spaces that are difficult to inspect or control. While Sparse Autoencoders (SAEs) can decompose hidden activations into more fine-grained, interpretable features, they often struggle to reliably align these features with human-defined concepts, resulting in entangled and distributed feature representations. To address this, we introduce ALIGNSAE, a method that aligns SAE features with a predefined ontology through a "pre-train, then post-train" curriculum. After an initial unsupervised training phase, we apply supervised post-training to bind specific concepts to dedicated latent slots while preserving the remaining capacity for general reconstruction. This separation creates an interpretable interface where specific concepts can be inspected and controlled without interference from unrelated features. Empirical results on GPT-2 across synthetic biographical QA (1-hop, 6 concepts) and compositional relational reasoning (2-hop, 20 concepts) demonstrate that AlignSAE enables precise causal interventions, such as reliable concept swaps (94.8% success in 1-hop), by targeting single, semantically aligned slots. AlignSAE achieves disentanglement improvements of $9-\infty\times$ over standard SAEs (measured via Mutual Information Gap), and further supports multi-hop reasoning and a mechanistic probe of grokking-like generalization dynamics. We evaluate on a single model architecture (GPT-2); the method itself is model-agnostic.

---

[†] Corresponding authors.
[‡] Work done while MY was a student at the University of Arizona.

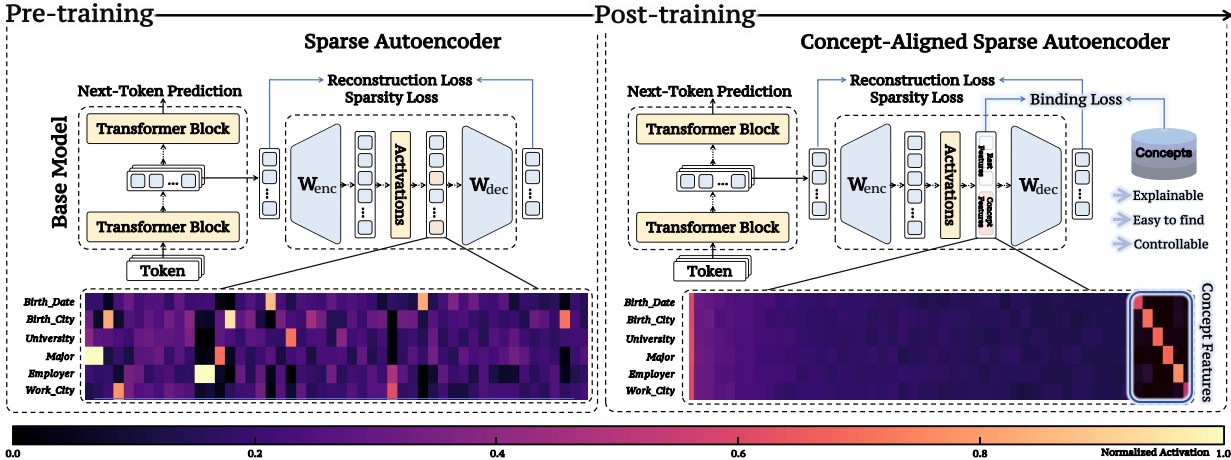

Figure 1: Overview of ALIGNSAE. Left: An unsupervised SAE trained post hoc on frozen LLM activations optimizes only reconstruction and sparsity, so each concept tends to be spread across multiple features, making interventions unreliable. Right: Our ALIGNSAE adds a supervised binding loss that maps each concept in a pre-defined ontology to a dedicated feature, yielding clean, isolated activations that are easy to find, interpret, and control.

# 1 Introduction

While Large Language Models (LLMs) have advanced rapidly, their internal mechanisms remain opaque. Mechanistic interpretability (Bereska & Gavves, 2024; Huben et al., 2024) seeks to bridge this gap by reverse-engineering these models and decomposing their internal computations into interpretable components. Early efforts focused on inspecting individual neurons (Nguyen et al., 2019; Elhage et al., 2022; Bills et al., 2023), operating under the assumption that specific neurons would map one-to-one onto human concepts. However, this approach faced a fundamental barrier of superposition, *i.e.*, neural networks represent more independent features than available neurons by encoding each feature as a linear combination of neurons (Ferrando et al., 2024). Consequently, single neurons become difficult to interpret, as their activations represent entangled mixtures of concepts.

This limitation of neuron-level analysis directly motivated the development of *Sparse Autoencoders (SAEs)*. The idea is to disentangle these superimposed neurons into more interpretable *features*, by learning an overcomplete, sparse representation of neural activations (Shu et al., 2025). By mapping LLM's hidden states into higher-dimensional space, SAEs often learned features that are cleaner and more interpretable than individual neurons (Leask et al., 2025; Chanin et al., 2025; Yan et al., 2025). Ideally, features learned by an SAE would correspond to atomic, independent, human-interpretable concepts, so that a human can easily inspect, interpret, and steer model behavior. For example, in the context of relation extraction, one would expect a single SAE feature to exclusively represent the relation BIRTH_CITY, such that manipulating this feature alone would precisely control the model's output regarding cities of birth. However, because standard SAEs are trained in an unsupervised fashion, they *have no explicit incentive to align their latent features with human-defined concepts*. In practice, this leads to two major challenges in interpreting the SAE feature space: 1) *Feature interpretation is non-trivial*: determining which feature corresponds to a target concept is difficult. To infer a feature's semantics, practitioners often rely on constructing minimal contrast pairs (Jing et al., 2025; Li et al., 2025) or inspecting top-activating examples (Cunningham et al., 2023; Bereska & Gavves, 2024; Shu et al., 2025). 2) *Features remain entangled*: concepts are often fragmented across multiple features, and conversely, a single feature may correspond to multiple unrelated concepts, as shown in Fig. 1 (left). These limitations undermine downstream applications that require reliable feature-level control, such as safety steering (Bereska & Gavves, 2024; Bhattacharjee et al., 2024; Ghosh et al., 2025; O'Brien et al., 2025), knowledge editing (Makelov et al., 2024; Guo et al., 2024; Farrell et al., 2024; Zhao et al., 2025; Karvonen et al., 2025), and data attribution (He et al., 2024; Muhamed et al., 2025).

To mitigate these issues, we take inspiration from the training pipeline of LLMs. As illustrated in Fig. 1, we view conventional SAE training as analogous to LLM pre-training: an unsupervised phase that discovers a broad latent feature space but does not guarantee alignment with human concepts. In LLMs, this misalignment is addressed by post-training steps such as instruction tuning (Wei et al., 2022; Zhang et al., 2025) or RLHF (Ouyang et al., 2022). By analogy, we propose an **SAE post-training** stage that guides the unsupervised SAE to align the SAE's feature space with a set of chosen concepts, turning it from a reconstructive tool into a reliable concept-level interface.

Concretely, we attach an SAE to one layer of a frozen base LM and train it in two phases: first unsupervised, then supervised. After the SAE has learned to reconstruct the activations, we "fine-tune" the SAE with concept supervision. We reserve $|\mathcal{R}|$ dedicated concept slots (one per ontology concept), while the remaining $K$ dimensions form a free feature bank to preserve reconstruction fidelity. We then augment the training objective with additional losses to bind and isolate each concept in its corresponding feature slot. In particular, we impose: (i) a *concept binding loss* that forces a one-to-one mapping between each labeled concept and a dedicated feature, (ii) a *concept invariance loss* that makes each concept feature invariant to irrelevant variations and decorrelates it from the free features, and (iii) a *sufficiency loss* that trains an auxiliary answer head to rely only on the concept slots for predicting concept-related information. Together, these objectives encourage the encoder to route concept-specific evidence into the appropriate slot rather than dispersing it across the latent space. The result is an SAE feature space that directly corresponds to human-interpretable concepts, as shown in Fig. 1 (right). In this work, we focus on the important NLP task of relation completion (RC). Accordingly, we use an ontology where concepts correspond to relation types, *e.g.*, BIRTH_CITY.[1]

Empirically, ALIGNSAE is more interpretable, disentangled, and controllable than a standard SAE under three RC evaluations. **(i) Binding/generalization:** concept–slot binding becomes clean and diagonal at mid layers (Fig. 3) and transfers to unseen templates (Fig. 4). **(ii) Disentanglement:** concepts concentrate onto single slots with low fragmentation and high Top-1 mass (Fig. 5). **(iii) Causal control:** swap interventions exhibit a clear fidelity–strength tradeoff (Table 1), strong mid-layer controllability (Fig. 6), and predictable failure modes (Table 2); in 2-hop RC, post-trained ALIGNSAE attains 4× higher swap success than the traditional SAE (Fig. 7).

## 2 Related Work

We review two main directions that influenced this work: *Sparse Autoencoder Steering* and *Concept Binding*, which address interpretable control and concept alignment, respectively. Inspired by both, we introduce ALIGNSAE, a lightweight, concept-aligned, and interpretable SAE framework.

**Sparse Autoencoder Steering.** Sparse Autoencoders (SAEs) provide an interpretable interface to LLM activations by decomposing superposed, polysemantic neuron activity into sparse, overcomplete features (Bricken et al., 2023; Cunningham et al., 2023). This representation enables *SAE steering*, where intervening on specific features can causally influence model outputs toward desired behaviors or concepts (O'Brien et al., 2025; Marks et al., 2025; Arad et al., 2025). Recent work improves feature quality through training objectives and architectural choices (Rajamanoharan et al., 2024; Bricken et al., 2023; Shu et al., 2025; Sharkey et al., 2025). However, since SAE features are learned in a purely unsupervised manner, they are not guaranteed to align with a user-specified concept set: a target concept may be fragmented across multiple features, and individual features may mix unrelated signals. As a result, practical steering often still relies on manual feature identification, including contrast pairs and feature search heuristics (O'Brien et al., 2025; Jing et al., 2025; Bayat et al., 2025; Chalnev et al., 2024; Yang et al., 2025), which are limited due to the remaining poly-semanticity of the features (Chanin et al., 2025; Cui et al., 2025; Minegishi et al., 2025). To address this, we post-train an SAE with concept supervision to learn a stable concept–feature mapping: we allocate dedicated concept slots, bind each ontology relation to a fixed slot, and make steering interventions directly targetable without post-hoc feature search. A concurrent work, G-SAE (Härle et al., 2025), also conditions SAE latent representations on labeled concepts to enforce localized, monosemantic features—the fact that two independent groups arrive at this core insight strengthens the case for concept-supervised SAE training. AlignSAE goes further in several respects: (i) we demonstrate *causal controllability* via concept swaps (94.8%

---

[1]In principle, ALIGNSAE should be applicable to any type of ontology. We leave this additional study as future work.

success in 1-hop), a stronger test than G-SAE's feature-level monosemanticity scores (FMS); (ii) we evaluate on *compositional multi-hop reasoning*, a setting G-SAE does not address; (iii) our value heads provide a *sufficiency guarantee*—each concept slot predicts the concept's actual value, not just its label; and (iv) we reveal a grokking-like phase transition in binding emergence, providing mechanistic insight into the alignment process.

**Concept Binding.** Posterior Regularization (PR) (Ganchev et al., 2010) and Logic Rule Encoding (LRE) (Hu et al., 2016) are traditional frameworks adopted to bind human-defined concepts to neural models by imposing soft constraints on posterior distribution, or integrating logic rules into the learning objectives. Prior work has applied PR to reading comprehension by enforcing linguistic concept-level constraints (Zhou et al., 2019), and to question answering by mapping event triggers to conceptual constraints (Lu et al., 2023). LRE works (Hu et al., 2016; Fischer et al., 2019; Yang et al., 2023) reconstruct the training objective by combining the task loss with a rule loss, thereby binding logical concepts to model predictions via parameter updates. Although LRE addresses the soft-constraint issue of PR, it remains a black-box mechanism that cannot be used to interpret or control specific internal representations of the model. Concept Bottleneck Models (Koh et al., 2020), originally proposed for image classification, map intermediate representations to human-readable concepts via a supervised concept-mapping loss. However, such heavy-weight intervention into the base model architecture is not easily scalable to large-scale models. To address these gaps, we propose a lightweight and interpretable framework, ALIGNSAE, which binds human-readable concepts to the intermediate representations of a frozen base model.

**Disentangled Representation Learning.** AlignSAE draws on principles from the disentanglement literature. $\beta$-VAE (Higgins et al., 2017) and FactorVAE (Kim & Mnih, 2018) use modified ELBO objectives to encourage factorial latent distributions; our alignment loss serves an analogous purpose but with *explicit* concept supervision rather than unsupervised statistical independence. We adopt the Mutual Information Gap (MIG) (Chen et al., 2018) as a quantitative disentanglement metric (§G). Concept Bottleneck Models (Koh et al., 2020) create interpretable intermediate representations aligned with human-readable concepts; AlignSAE can be viewed as a post-hoc concept bottleneck applied to frozen model activations, without modifying the base model architecture. A key distinction from classical disentanglement is that standard methods assume *statistically independent* factors of variation—an assumption that often fails for language (e.g., employer and work_city are correlated). Our experiments confirm this: a standard SAE trained with only reconstruction and sparsity achieves MIG of exactly 0.000 for all 1-hop concepts, because without explicit supervision, no single feature has any incentive to exclusively capture a specific concept. AlignSAE's supervised binding addresses this directly.

## 3  Method

We construct an explicit, verifiable, and controllable interface over a frozen LLM's activations via a Concept-Aligned Sparse Autoencoder (AlignSAE). This interface provides evidence that the model encodes concept-relevant information in a linearly accessible form, though the one-to-one slot mapping is a product of the supervised training objective rather than necessarily reflecting the model's native representational structure.

**Terminology.** In this work, we focus on the task of relation completion (RC), where facts are represented as triples $(e_1, r, e_2)$: the relation type $r$ links entity mentions $e_1$ and $e_2$ (*e.g.*, (*"Marie Curie"*, BORN_IN, *"Warsaw"*)). In this paper, RC appears as *two tasks*. For 1-hop, given $(e_1, r)$, the model predicts the missing object $e_2$; for instance, *"What is Alice's* birth date*?"* corresponds to a single relation query (§4.2.1). For 2-hop, given $(e_1, r_1, r_2)$, the model predicts $e_3$ by composing two relations through an intermediate entity $e_2$, *i.e.*, $e_1 \xrightarrow{r_1} e_2 \xrightarrow{r_2} e_3$; *e.g.*, *"Who is the* classmate *of the* mentor *of Barbara?"* (§4.2.2). Here we instantiate concepts as relation types from a domain ontology, a finite inventory of semantic links between entities (*e.g.*, BORN_IN, FRIEND_OF). We focus on relations because RC is a core NLP task with high-impact closed-domain applications (*e.g.*, biomedicine (Demner-Fushman et al., 2025), finance (Grishman & Sundheim, 1996)), where the target relations are defined by an ontology; nevertheless, our framework is not tied to relations and could bind other ontology concepts (*e.g.*, entities or attributes) to supervised SAE slots.

**Concept Binding in Activation Space.** Given a frozen LLM $\mathcal{M}$, we extract an activation $h \in \mathbb{R}^d$ from layer $\ell$ and learn a sparse representation $z = E(h) \in \mathbb{R}^{|\mathcal{R}|+K}$. We partition $z = [z_{\text{rest}}; z_{\text{concept}}]$, where $z_{\text{concept}} \in \mathbb{R}^{|\mathcal{R}|}$ are the supervised concept slots and $z_{\text{rest}} \in \mathbb{R}^K$ are the free features ($|\mathcal{R}| \ll K$). We define the average activation of concept $k$ on slot $j$ as: $\bar{a}_{k,j} = \mathbb{E}_{x \sim \mathcal{D}_k} \left[ z_j^{(k)}(x) \right]$, where $\mathcal{D}_k$ denotes the subset of inputs labeled with concept $k$, and $z_j^{(k)}(x)$ is the activation of slot $j$ for input $x$ with concept label $k$. The supervised slots are indexed as $j \in \{K+1, \dots, K+|\mathcal{R}|\}$. We supervise $z_{\text{concept}}$ so that each ontology concept occupies its own dedicated slot: for an input containing concept $c$, its slot should activate while other concept slots are suppressed. This makes concept–slot activations a direct, verifiable readout of the concept in use. We keep $z_{\text{rest}}$ "free" to absorb residual linguistic variation and keep the base model's statistical structure.

**A Verifiable Interface.** Unlike a traditional SAE, which is primarily diagnostic, our approach constructs an operational interface that can be validated by intervention rather than relying on minimal comparison pairs. Specifically, (i) *verification*: we can check whether the model is using a particular relation by observing whether the corresponding concept slot activates; and (ii) *control*: we can causally steer the computation by manually activating or suppressing that slot, directly influencing the model's downstream prediction.

**Model Training.** Although AlignSAE is LLM-agnostic, we evaluate it on GPT-2.[2] We fine-tune the base model on our RC training data (§B), extract layer activations at the final question token across all layers, and train AlignSAE with a task-dependent free-feature bank of size $K$ and $|\mathcal{R}|$ concept slots using a two-stage training (§C, 4.1).

**SAE Pre-Training and Post-Training.** Directly optimizing the full objective from scratch can produce unstable binding (§6.2), so we adopt a two-phase curriculum that parallels LLM pre-/post-training (Fig. 1). In the *pre-training* phase, the SAE is trained on reconstruction and sparsity, allowing the decoder to form a high-capacity dictionary. In subsequent *post-training*, we strengthen the binding and value losses, then activate the orthogonality penalty. This reshapes the latent space so supervised slots become clean, disentangled carriers of atomic concepts, while remaining decoupled from the free feature bank. This curriculum retains the benefits of joint optimization (concept slots that are simultaneously interpretable and task-predictive) while avoiding the degenerate minima that arise when strong supervision is applied before the underlying representation has stabilized (§C).

**Objectives.** We train the encoder, decoder, and value head jointly. Our binding objective augments the standard SAE loss ($\mathcal{L}_{\text{SAE}}$) with (i) a supervised loss that assigns each relation to a dedicated concept slot ($\mathcal{L}_{\text{align}}$), (ii) a $z_{\text{concept}}-z_{\text{rest}}$ decorrelation penalty to reduce concept leakage into $z_{\text{rest}}$ ($\mathcal{L}_\perp$), and (iii) an auxiliary value loss that encourages the concept slots to support answer prediction ($\mathcal{L}_{\text{val}}$) (Agarwal et al., 2021; Hewitt et al., 2023; Sun et al., 2024), where $y_{\text{ans}}$ is the ground-truth answer:

$$
\begin{aligned}
\mathcal{L}_{\text{SAE}} &= \lambda_{\text{rec}} \|h - \hat{h}\|_2^2 + \lambda_{\text{sp}} \|z\|_1, \\
\mathcal{L}_{\text{align}} &= \text{CE}\big(\text{softmax}(z_{\text{concept}}), y_{\text{rel}}\big), \\
\mathcal{L}_\perp &= \|\text{corr}(z_{\text{concept}}, z_{\text{rest}})\|_F^2, \\
\mathcal{L}_{\text{val}} &= \text{CE}\big(\text{softmax}(V(z_{\text{concept}})), y_{\text{ans}}\big), \\
\mathcal{L} &= \mathcal{L}_{\text{SAE}} + \lambda_{\text{align}}\mathcal{L}_{\text{align}} + \lambda_\perp \mathcal{L}_\perp + \lambda_{\text{val}}\mathcal{L}_{\text{val}}.
\end{aligned}
\tag{1}
$$

See §C.3 for full definitions and hyperparameters. We also assess each term through ablations in §6.

Overall, our method turns a frozen LLM's conceptual evidence into an explicit, verifiable, and controllable interface by (i) training an SAE on a single intermediate layer, (ii) allocating one-to-one ontology-aligned concept slots, and (iii) using a two-stage curriculum that yields disentangled, slot-addressable features. Because only the lightweight SAE is trained, the base model remains fixed, enabling efficient diagnosis (slot verification) and causal control (slot interventions).

---

[2]We choose GPT-2 for controlled analysis under compute limits, enabling layer-wise probing, ablation/steering analysis.

# 4 Implementation

We attach the SAE to a frozen base LLM (*e.g.*, GPT–2; our approach is not limited to a specific model) by extracting a representation $h \in \mathbb{R}^d$ from an intermediate layer $\ell$. Concretely, given an input biography question $x$, the model produces token-level hidden states at each layer, and we pool the activations from layer $\ell$ (*e.g.*, mean pooling) to form $h$. The base LLM is kept frozen, preserving its linguistic competence while placing all supervision on the light-weight SAE interface (§B.4).

## 4.1 Concept-Aligned Sparse Autoencoder

We use a supervised SAE that exposes an interpretable control surface aligned with $\mathcal{R}$, while delegating residual variance to a large bank of free features. The encoder $E : \mathbb{R}^d \to \mathbb{R}^{|\mathcal{R}|+K}$ maps $h$ to a sparse code $z = \mathrm{ReLU}(W_e h + b_e)$, partitioned as $z = [z_{\mathrm{concept}}; z_{\mathrm{rest}}]$ with $z_{\mathrm{rest}} \in \mathbb{R}^K$ (unsupervised free features) and $z_{\mathrm{concept}} \in \mathbb{R}^{|\mathcal{R}|}$ (supervised concept slots). The decoder $D : \mathbb{R}^{|\mathcal{R}|+K} \to \mathbb{R}^d$ reconstructs $\hat{h} = W_d z + b_d$. We set $K$=10,000 (1-hop) and $K$=100,000 (2-hop) to add capacity without burdening the $|\mathcal{R}|$ interpretable slots, read out concept evidence from $z_{\mathrm{concept}}$ at the output token, and post-train the SAE with the loss in §3.

## 4.2 Tasks and Datasets

We evaluate our approach ALIGNSAE on two controlled benchmarks designed to test both direct attribute retrieval and multi-step logical derivation: (i) *biography factual recall* (1-hop) and (ii) *multi-step compositional reasoning* (2-hop).

### 4.2.1 Factual Recall (1-hop)

We validate our approach on a biography QA task over a fixed ontology. Let $\mathcal{R}$ be a fixed ontology of $|\mathcal{R}| = 6$ (Table 6; §A). For a person $p$ and relation $r \in \mathcal{R}$, a canonical table provides the gold value $y^\star = g(p, r)$ (*e.g.*, *Wesleyan University* for UNIVERSITY). Each input $x$ is a natural-language question that mentions $p$ and implicitly targets a single relation $r^\star \in \mathcal{R}$; the model must generate the corresponding value $y^\star = g(p, r^\star)$.

We adopt the synthetic biography dataset of Allen-Zhu & Li (2024) with minor modifications to better separate *semantic binding* from *template memorization*. We generate 1,000 person profiles (five biography variants each) and instantiate questions from paraphrase templates that preserve the same underlying relation. To assess robustness to surface form, we use a disjoint template split: we train on all profiles using two templates and evaluate UNSEEN-TEMPLATE generalization on two held-out templates (Table 6; §A). The held-out templates use different lexical triggers (*e.g., born* vs. *birth city, alma mater* vs. *university*) to reduce n-gram overlap with training prompts. We also explored *LLM-generated questions* by few-shot prompting Claude 3.5 Sonnet on 1-hop RC queries ($e_1$,$r$) to generate instance-specific paraphrases, but excluded them since many are too indirect for base GPT-2 (details in Fig. 10; §A.4).

### 4.2.2 Compositional Reasoning (2-hop)

We further evaluate ALIGNSAE on a *two-hop* compositional reasoning task (Du et al., 2025). We use 2-hop as a compositional concept test for ALIGN-SAE. Unlike 1-hop, it decomposes prediction into two concept steps with an intermediate entity $e_2$, so concept–slot binding can be *verified* and *intervened* at the step where each concept is actually used, rather than inferred only from final-answer accuracy. This decomposition also enables targeted causal tests (*e.g.*, swapping the $r_2$ at the second step), making controllability more diagnostic. Each example provides a start entity $e_1$ and relations ($r_1$,$r_2$); the model must infer $e_1 \xrightarrow{r_1} e_2 \xrightarrow{r_2} e_3$ (*i.e.*, $e_2$=$r_1(e_1)$, $e_3$=$r_2(e_2)$).

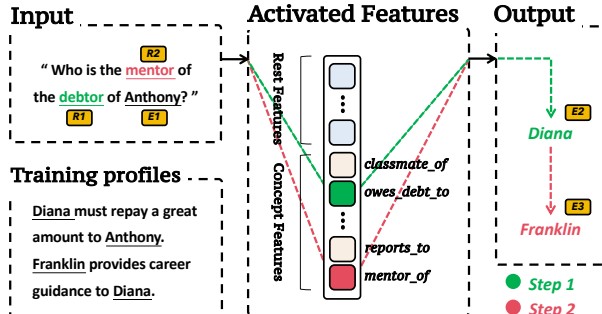

Figure 2: 2-hop step-wise alignment. With ($e_1$,$r_1$,$r_2$), the model predicts ($e_2$,$e_3$); $r_1$ activates at $e_2$ and $r_2$ at $e_3$, with remaining capacity in unsupervised features.

Inputs use templates like *"Who is the $r_2$ of the $r_1$ of $e_1$?"* with brief profile sentences stating the two hop facts (Fig. 17; §M). The dataset instantiates an ontology of 20 relations over 60 entities (see §M). Following Fig. 2, we train the model to output "$e_2; e_3$" and apply step-wise supervision: the $r_1$ slot should activate when generating $e_2$, and the $r_2$ slot when generating $e_3$.

### 4.3 Metrics

To measure concept–slot alignment, we build a relation–slot confusion matrix $C$ where $C[r,j]$ is the normalized fraction of examples whose top-activated slot at the answer token is $j$ for gold relation $r$, and report $\text{Acc}_{\text{bind}} := \frac{1}{|\mathcal{R}|} \sum_{r \in \mathcal{R}} C[r,r]$.

### 4.4 Controllability (Swap Test)

To evaluate *causal* controllability, we perform a *swap test*: given a query whose gold relation is $r^\star$, we intervene by injecting the decoded direction of an alternative relation slot $j \neq r^\star$ with strength $\alpha > 0$. Formally, letting $h_{\ell,t} \in \mathbb{R}^d$ denote the residual stream at layer $\ell$ and token position $t$, and $W_{\text{dec}} \in \mathbb{R}^{d \times K}$ denote the SAE decoder with reconstruction $\hat{h}_{\ell,t} = W_{\text{dec}}z_{\ell,t}$, the decoded basis direction for slot $j$ is $v_j := W_{\text{dec}}e_j$ and the intervention is $h'_{\ell,t} = h_{\ell,t} + \alpha v_j = h_{\ell,t} + W_{\text{dec}}(\alpha e_j)$. We count a swap as successful if the model's top-1 generated answer matches the gold value under the intervened relation, *i.e.*, $g(p,j)$, rather than the original target $g(p,r^\star)$ (full definitions in §D).

## 5 Results: Factual Recall (1-hop)

We first evaluate the proposed interface across transformer layers, data (Train, Test–Unseen), and controllability settings. Metrics follow §4.3.

### 5.1 Layer-Wise Performance

The contrast between layer 0 and layer 6 illustrates how concept alignment emerges (Fig. 3). At layer 0, the relation–slot confusion matrix is diffuse with substantial off-diagonal mass and overlapping activations, indicating entangled features that fail to isolate ontology relations. At layer 6, the same matrix collapses to a sharp diagonal with negligible off-diagonal mass, yielding a stable permutation that assigns each relation to a single slot.

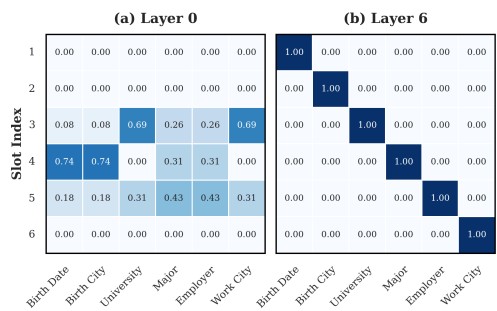

Figure 3: Concept–slot binding at a shallow layer (a) versus a mid layer (b). At layer 0, supervision for each relation is dispersed across multiple slots, whereas at layer 6 the SAE learns a perfect one-to-one binding.

Table 1 and Fig. 4 reveal a clear mid-layer sweet spot. Performance peaks around Layer 6, where binding is essentially one-to-one (*Diagonal Acc*= 1.00), swap controllability is highest, and paraphrase generalization remains strong (*Test Unseen Acc*= 0.912). In contrast, early layers are poorly aligned (Layer 0, *Diagonal Acc*= 0.238), with diffuse off-diagonal mass (Fig. 3). Although Layer 6 incurs higher reconstruction error, it yields a large gain in controllability (*swap success* +0.81), suggesting that clean semantic binding emerges in mid layers with richer contextual representations, while early embedding-dominated layers and deeper, more compressed layers make a stable slot interface harder to preserve.

### 5.2 Concept–Feature Alignment

To quantify how cleanly each concept is represented at different depths, we compare a traditional SAE trained purely unsupervised with ALIGNSAE, which adds concept-supervision as a post-training signal. Let $z_i \in$

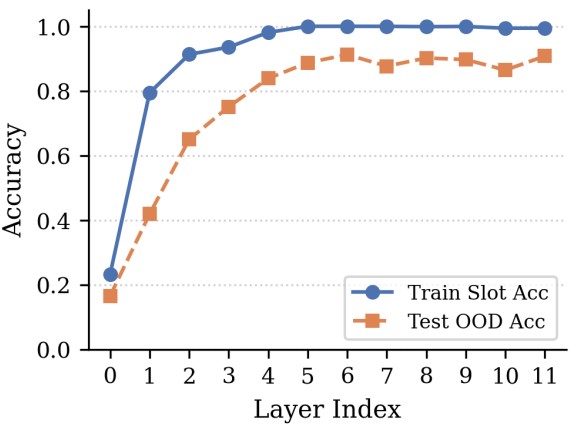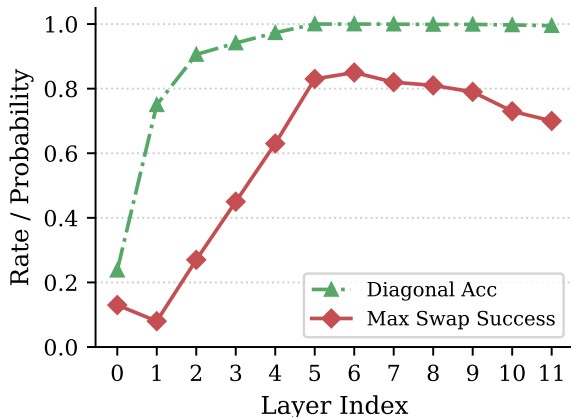

Figure 4: Binding generalization vs. causal control, showing performance on train and unseen distributions (left) alongside the corresponding diagonal and swap mechanisms (right).

$\mathbb{R}^{|\mathcal{R}|+K}$ be the sparse code for example $i$, and let $c(i)$ be its concept label (*e.g.*, one of six ontological relations). We first define the *average activation* of concept $c$ on feature $k \in \{1, \ldots, |\mathcal{R}|\}$ as $A_{c,k} = \mathbb{E}_{i:c(i)=c}\left[z^{(i)}_{\text{concept},k}\right]$, and normalize over features to obtain a concept–feature distribution $B_{c,k} = \frac{A_{c,k}}{\sum_{k'} A_{c,k'} + \epsilon}$. From $B_{c,\cdot}$ we derive two summary metrics for each concept $c$:

**Effective number of features (EffFeat).** We measure how many features are effectively used to represent a concept via the entropy of $B_{c,\cdot}$: $\text{EffFeat}(c) = \exp\left(-\sum_k B_{c,k} \log B_{c,k}\right)$, where smaller values indicate that a concept is concentrated on fewer features (*i.e.*, lower fragmentation).

**Top-1 concentration (Top1C).** We also track how much of a concept's mass is captured by its single most responsive feature: $\text{Top1C}(c) = \max_k B_{c,k}$, where larger values indicate that one feature dominates the representation of concept $c$.

| Metric | Layer 0 | Layer 6 | $\Delta$ (L6 $-$ L0) |
|---|---|---|---|
| Diagonal Acc ↑ | 0.238 | **1.000** | ↑ 0.76 |
| Swap Success ↑ | 0.040 | **0.847** | ↑ 0.81 |
| Train Slot Acc ↑ | 0.232 | **1.000** | ↑ 0.77 |
| Test Unseen Acc ↑ | 0.165 | **0.912** | ↑ 0.75 |
| Recon MSE ↓ | $6.53 \cdot 10^{-5}$ | $7.42 \cdot 10^{-2}$ | ↑ $\approx 1.1\mathbf{k}\times$ |

Table 1: Layer 0 vs. Layer 6 performance comparison.

Fig. 5 reports layer-wise averages of these metrics over six concepts for ALIGNSAE and a traditional SAE. The traditional SAE (pre-training only) remains highly fragmented across all layers: EffFeat spans hundreds to thousands of features per concept and Top1C $\approx 0$, suggesting that concepts are not consistently localized to a single feature. In contrast, ALIGNSAE sharply reduces fragmentation (EffFeat $\approx 1$) and increases concentration via post-training supervision, yielding compact representations from mid layers onward and near one-to-one bindings in deeper layers. Overall, concept-level post-training converts a diffuse many-to-many feature space into a compact, interpretable interface with directly addressable concepts.

Detailed disentanglement and robustness analyses are deferred to Sections G to I.

## 5.3 Swap Controllability

We probe whether concept slots behave as usable control knobs by measuring swap success under intervention (see §4.4). Fig. 6 reports average swap success rate (average over 1,000 examples) across layers and amplification strengths: moderate amplification ($\alpha \approx 2$) reliably switches the answer type at mid layers (5–8). At layer 6, suc-

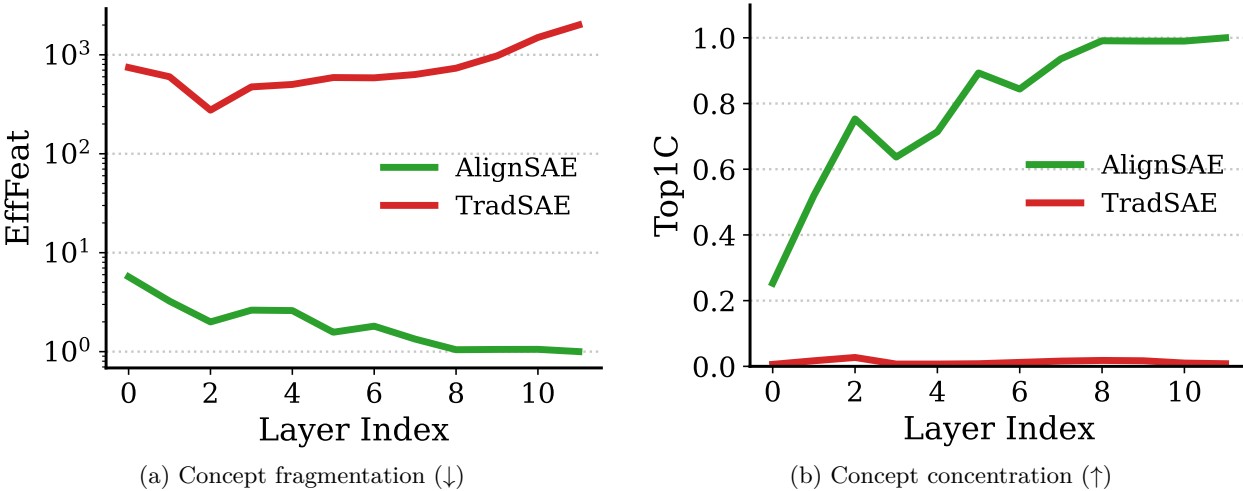

(a) Concept fragmentation (↓)    (b) Concept concentration (↑)

Figure 5: Layer-wise fragmentation (↓) and concentration (↑) for ALIGNSAE and a traditional SAE.

cess rises from 0.54 at $\alpha$=1.0 to 0.85 at $\alpha$=2.0, but drops to 0.23 at $\alpha$=10.0, indicating that over-amplification destabilizes the intervention and revealing a narrow regime where control is effective and predictable.

For example, for a BIRTH_DATE question, the model originally answers *"24, March, 1964"*; amplifying the UNIVERSITY slot with $\alpha$=2 switches the output to *Wesleyan University*, showing that slots are causal control handles rather than merely diagnostic (more examples in Table 7 & Fig. 12; §F.2). We further ablate the post-training objectives; results in §6 show each component is necessary for robust controllability. Overall, early layers are too local and deeper layers too compressed; mid-layer post-training (*e.g.*, Layer 6) yields near-perfect binding and reliable swaps at $\alpha \approx 2$, the practical regime for controllable concept access. Detailed side-effects evaluation (perplexity under correct/wrong-slot steering and LLM-judge free-form quality) is reported in §I.

### 5.4 Swap Error Analysis

Even when swap steering fails to hit the *exact* gold entity, it often preserves the *answer category* of the intended swapped concept. Table 2 reports *Category Preservation*, computed *only over failed swaps*. Under moderate amplification ($\alpha$=2), 75.3% of failures remain in the correct category; under strong amplification ($\alpha$=10), this rises to 83.0%, suggesting that larger interventions more reliably move the model onto the right target type even as exact entity selection degrades.

For example, in Table 3, swapping UNIVERSITY→MAJOR at $\alpha$=10 yields a *type*

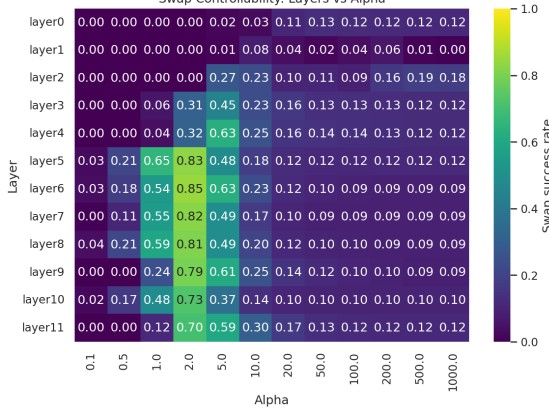

Figure 6: Swap controllability across layers and amplification $\alpha$. Mid layers remain robust at $\alpha \approx 2$.

| Target swap | $\alpha$=2 (15% Err) | | | $\alpha$=10 (77% Err) | | |
|---|---|---|---|---|---|---|
| | S | D | S% | S | D | S% |
| birth_city | 36 | 20 | 64.3 | 91 | 50 | 64.5 |
| birth_date | 20 | 0 | 100.0 | 139 | 0 | 100.0 |
| employer | 8 | 1 | 88.9 | 134 | 3 | 97.8 |
| major | 1 | 0 | 100.0 | 77 | 51 | 60.2 |
| university | 42 | 10 | 80.8 | 101 | 1 | 99.0 |
| work_city | 9 | 7 | 56.2 | 94 | 25 | 79.0 |
| **Overall** | **116** | **38** | **75.3** | **636** | **130** | **83.0** |

Table 2: Category retention on *failed* swaps at Layer 6. "Same" (S) means correct semantic class; "Diff" (D) is outside.

| Swap | **Q:** Where did Jesse Kian Tate go to college? |
|------|--------------------------------------------------|
| | UNIV → **Swap:** MAJOR |

| Outputs | **Base:** Rochester Inst. of Tech. | |
|---------|-------------------------------------|---|
| | **Gold:** Physical Therapy | |
| | **Gen: Geography** | [ Type ✔ Entity ✘ ] |

Table 3: A failure case: steering ($\alpha = 10$) successfully flips the semantic category (type) but fails to retrieve the specific ground-truth entity.

| Config | Train Slot | Test-OOD Slot | Train Acc | Test-Unseen Acc | Swap Success Rate |
|--------|-----------|---------------|-----------|-----------------|-------------------|
| **joint** | **1.000** | **0.912** | **1.000** | **0.809** | **0.847** |
| no_align | 0.167 | 0.167 | **1.000** | **0.809** | 0.053 |
| no_ortho | 0.711 | 0.689 | **1.000** | **0.809** | 0.046 |
| no_value | 0.177 | 0.172 | **1.000** | **0.809** | 0.720 |

Table 4: **Ablations of AlignSAE post-training. Best** configuration is **joint**, which achieves strong OOD binding and high swap success simultaneously. Swap success is reported at each configuration's best-performing intervention strength ($\alpha$: joint=2, no_align=0.1, no_ortho=10, no_value=100). Chance slot binding for 6 relations is $1/6 \approx 0.167$.

✓ but *entity* ✗ answer: *Geography* instead of *Physical Therapy*. In this controlled setting, our strict swap metric may understate controllability: even without exact entity matches, interventions often enforce the target concept.

## 6 Ablation Study

### 6.1 Effect of Training Objectives

We ablate three post-training components of ALIGNSAE: (i) binding/alignment ($\mathcal{L}_{\text{align}}$), (ii) orthogonality ($\mathcal{L}_{\perp}$), and (iii) value/sufficiency ($\mathcal{L}_{\text{val}}$), while keeping the underlying task and backbone fixed.

**Results.** As shown in Table 4, **binding is necessary for alignment:** removing $\mathcal{L}_{\text{align}}$ collapses slot binding to chance (0.167 Test-Unseen) and nearly eliminates swap control (0.053). **Orthogonality is necessary for control:** without $\mathcal{L}_{\perp}$, binding remains high (0.689 Test-Unseen) but swaps fail (0.046), indicating leakage that breaks causal interventions. **Value improves interface quality:** removing $\mathcal{L}_{\text{val}}$ yields near-chance binding (0.172 Test-Unseen) yet can still achieve moderate swap success (0.72) at very large $\alpha$, whereas the full model achieves both strong Test-Unseen binding (0.891) and high swap success (0.847) at a small $\alpha = 2$.

### 6.2 Effect of Stage-1 Pretraining

We ablate the unsupervised Stage 1 and train ALIGNSAE *from scratch* with the same post-training objective (*No Stage 1*), keeping all other settings fixed.

**Results.** As shown in Table 5, removing Stage 1 substantially worsens optimization and representation quality: total loss increases by 79.8% ($1.86 \rightarrow 3.34$), with higher reconstruction error ($L_{\text{recon}}$: +61.1%) and a large degradation in sparsity ($L_{\text{sparse}}$: +1185.2%). Orthogonality also worsens ($L_{\text{ortho}}$: +41.8%). In contrast, slot binding accuracy changes little (slot_acc: +1.7%). Overall, Stage 1 is crucial for a stable sparse reconstructive basis, even if concept binding remains feasible without it.

| Metric | No Stage 1 | With Stage 1 | $\Delta$ (With–No) | $\%\Delta$ |
|---|---|---|---|---|
| $\mathcal{L}\downarrow$ | 3.3438 | 1.8594 | $L\downarrow$  1.4844 | **-44.4%** |
| $L_{\text{recon}}\downarrow$ | 0.0412 | 0.0256 | $L\downarrow$  0.0156 | **-37.9%** |
| $L_{\text{sparse}}\downarrow$ | 0.2682 | 0.0209 | $L\downarrow$  0.2473 | **-92.2%** |
| $L_{\text{align}}\downarrow$ | 0.0605 | 0.0762 | $L\uparrow$  0.0157 | **+26.0%** |
| $L_{\text{ortho}}\downarrow$ | 0.5330 | 0.3760 | $L\downarrow$  0.1570 | **-29.5%** |
| $L_{\text{value}}\downarrow$ | 3.3169 | 3.4125 | $L\uparrow$  0.0956 | **+2.9%** |
| $\text{Acc}_{\text{bind}}$ (train)$\uparrow$ | 0.9700 | 1.0000 | $A\uparrow$  0.0300 | **+3.1%** |
| $\text{Acc}_{\text{bind}}\uparrow$ | 0.9699 | 0.9538 | $A\downarrow$  0.0161 | **-1.7%** |
| $L_{\text{indep}}\downarrow$ | 157.8025 | 4.7621 | $L\downarrow$  153.0404 | **-97.0%** |

Table 5: Stage-1 ablation. Deltas are computed as **With–No**; losses prefer $\downarrow$ and accuracies prefer $\uparrow$.

# 7 Results: Multi-Step Reasoning (2-hop)

Building on the layer sweep in Fig. 6, we use Layer 6 for all 2-hop experiments, as it provides the most reliable substrate for slot-level binding and causal control in the step-wise setting (§4.2.2).

## 7.1 Step-Wise Binding and Swap Controllability

ALIGNSAE achieves 100% step-wise binding accuracy in 2-hop: the concept–slot confusion matrix is perfectly diagonal at both the $e_2$ step (binding to $r_1$) and the $e_3$ step (binding to $r_2$), confirming clean compositional alignment.

We next evaluate if these aligned slots are causal control knobs. Following the intervention protocol in §4.4, we inject a decoded direction corresponding to a target relation slot with strength $\alpha$ during inference and measure whether the model's output switches to the entity implied by the swapped relation. We compare against a traditional SAE and a supervised linear probe (Rimsky et al., 2024). We train a linear probe to predict the relation label from the frozen hidden state using cross-entropy loss ($\sim 100\%$ concept-classification accuracy). For swapping, we treat each classifier row weight $w_r$ as a concept direction and intervene directly in hidden space via $h'_{\ell,t} = h_{\ell,t} - \alpha w_{r_{\text{input}}} + \alpha w_{r_{\text{target}}}$. As shown in Fig. 7, the supervised ALIGNSAE enables higher swap success than both the traditional SAE and linear probe, peaking near $\alpha \approx 50$. Detailed comparison against a standard SAE baseline and a fuller analysis of the 2-hop controllability gap are provided in §K.

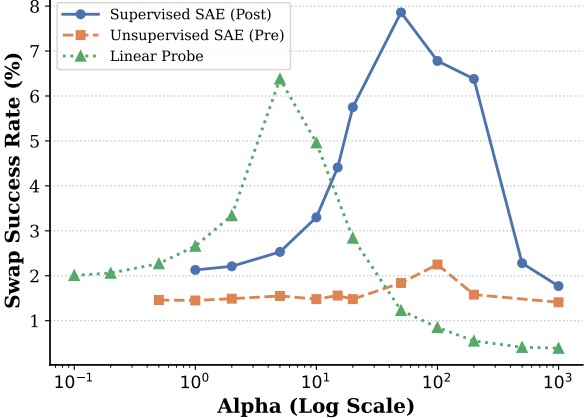

Figure 7: Swap controllability in 2-hop reasoning. The post-trained ALIGNSAE achieves $4\times$ higher swap success than the traditional SAE when $\alpha \approx 50$.

## 7.2 Grokking and Concept Binding

We use ALIGNSAE as a diagnostic probe to understand grokking in 2-hop reasoning (Wang et al., 2024; Ye et al., 2025). The base two-hop model exhibits grokking-like training dynamics: loss decreases smoothly, while validation accuracy remains low for many epochs before a sharp transition (Fig. 18 in §N).

Our hypothesis is that *before grokking*, concept evidence is present but not cleanly separated: the intended concept is encoded as a partially entangled mixture over slots. After the grokking transition, representations re-organize into structured, compositional features with stable step-wise 1-to-1 binding.

**Relation-to-Slot Binding Quality: Pre-Grokking vs Post-Grokking**

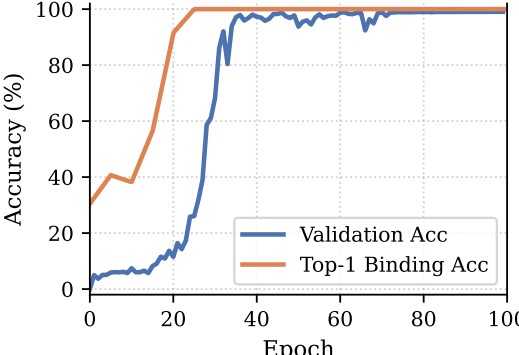

Figure 8: **Relation-to-slot binding (pre- vs. post-grokking).** At steps $e_2$ and $e_3$, binding is initially entangled but becomes perfectly diagonal after grokking, indicating clean step-wise alignment.

This predicts a *lag between knowing and showing.* In Fig. 9, Top-1 binding accuracy climbs rapidly and saturates early, whereas validation accuracy lags behind with a delayed jump. This dissociation suggests "hidden" progress: the model first organizes concept structure into stable internal features ("knowing"), and only subsequently learns to consistently *use* that structure for out-of-distribution generalization ("showing").

Fig. 8 visualizes this transition at Token 1 (the $e_2$ step). Pre-grokking, binding remains diffuse (epoch 10: 38.4%); post-grokking, the confusion matrix becomes perfectly diagonal (epoch 40: 100%). Together with the delayed accuracy jump, these results are consistent with grokking as a phase transition from entangled representations to a clean, compositional regime where step-specific concepts become slot-addressable features that support multi-step generalization.

Figure 9: **Grokking-like emergence.** Top-1 binding accuracy saturates early, while validation accuracy improves later, suggesting delayed generalization in 2-hop reasoning.

**Concurrent work.** Recent concurrent work also uses SAEs to study grokking/emergent transitions: Bereska et al. (2025) report sharp feature consolidation during grokking via an SAE-based superposition metric, while Kumar & Herlitz (2025) use SAE co-activation graphs and find no evidence that global topology forecasts emergent jumps. In contrast, we use *concept-aligned* slots as a *step-wise* diagnostic for how multi-hop evidence becomes compositional.

## 8 Conclusion and Future Work

In this work, we present ALIGNSAE, a framework that upgrades Sparse Autoencoders (SAEs) from descriptive probes to controllable interfaces. To mitigate feature entanglement in unsupervised SAEs, we introduce a "pre-train, then post-train" curriculum that enforces explicit alignment between designated latents and an external ontology, while retaining a large residual bank for reconstruction. This design yields reliable causal control, validated via paraphrase, robust concept swaps, and transfers cleanly to multi-hop reasoning. ALIGNSAE also enables fine-grained analysis of representation learning dynamics. We find a mechanistic account of grokking in compositional tasks: as generalization emerges, diffuse evidence concentrates into stable, step-wise bindings between relations and aligned slots. Furthermore, by grounding abstract concepts in latent structure, ALIGNSAE moves SAE-based interpretability toward controllable internal representations. A preliminary scalability study on FB15K (91 relations, §J) confirms that concept alignment remains effective beyond the original synthetic settings. Detailed applications and future directions are provided in Appendix L.

## Limitations

ALIGNSAE is evaluated in a deliberately controlled setting on a frozen GPT-2, using synthetic 1-hop biography QA and a 2-hop reasoning task rendered into natural-language prompts (§5, §7). This setup makes concept binding and causal interventions precisely measurable (§5.3) and allows us to extend grokking-style mechanistic analysis to the *concept-binding* level by tracking binding quality across training (Fig. 9 and Fig. 8). Unlike prior work on grokked transformers (Wang et al., 2024; Ye et al., 2025), which train on explicit symbolic IDs, we intentionally avoid entity/relation identifiers and instead use natural-language templates to better mimic realistic usage and reduce shortcut learning; this choice improves validity but also makes generalization harder and leaves open how our findings translate to ID-based settings. More broadly, our conclusions are currently limited to a GPT-2 backbone and templated data with paraphrase splits (§A.4); highly indirect LLM-generated question styles can exceed the backbone's QA capacity (UNSEEN-TEMPLATE generalization; §A.4), so we focus on unseen but direct template-level questions to ensure failures reflect representation-level alignment rather than backbone incapability. We also use a small ontology (*e.g.,* 20 relations in 2-hop), which does not reflect full-scale SAE deployments; scaling to larger ontologies/models is left to future work, though the proposed binding and causal evaluation are not tied to this size.

Additionally, we evaluate on a single model architecture (GPT-2) with synthetic data. While the controlled setting enables rigorous evaluation of binding, disentanglement, and controllability with unambiguous ground truth, it does not capture the full complexity of natural language. AlignSAE's design around a predefined ontology of discrete concepts is well-suited to domains with structured knowledge bases (*e.g.*, biomedicine, finance, law), where concepts are deterministic and well-defined. Extending to fuzzier or open-ended concept spaces—and to pre-trained models on natural corpora—is an important direction for future work.

## Broader Impact

Concept-aligned steering could in principle be misused to induce targeted biases or factual errors. However, the same capability is essential for *detecting and correcting* such biases. The transparency of named, inspectable concept slots is itself a safety advantage over opaque, distributed representations: interventions become auditable. Practical mitigations include: (i) AlignSAE requires model-internal access and labeled ontology data, limiting casual misuse; (ii) the slot interface can be monitored for unauthorized modifications; (iii) the framework can be used defensively—e.g., monitoring whether safety-relevant concept slots are being suppressed during inference.

## Acknowledgments

We thank the anonymous TMLR reviewers and the action editor for their valuable feedback. We gratefully acknowledge support from the University of Arizona Undergraduate Research Travel Grant, which provided funding for Minglai Yang. We also thank the College of Information Science for additional student research funding and the AI Club at University of Arizona for their support.

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

## Content of Appendix

## A  Biography Dataset Generation

### A.1  Synthetic Biography Dataset

We generated 1,000 synthetic person profiles, each containing six factual attributes: birth date, birth city, university, major, employer, and work city. Each person was paired with 5 biography variants constructed from template-based generation. For question-answering evaluation, we employed a template-split strategy: templates 0–1 were designated for training (in-distribution), while templates 2–3 served as out-of-distribution test cases to evaluate semantic generalization beyond pattern matching.

### A.2  Entity Vocabulary

The dataset drew from the following entity sets:

- **First names:** 411 diverse given names spanning traditional and modern choices
- **Middle names:** 461 names used for middle name generation
- **Last names:** 1,002 surnames representing common American family names
- **Birth cities:** 50 major U.S. cities (*e.g.*, New York, Los Angeles, Chicago, etc.)
- **Universities:** 341 U.S. colleges and universities spanning liberal arts colleges, research universities, technical institutes, and military academies
- **Majors:** 101 academic fields ranging from STEM disciplines (Computer Science, Mechanical Engineering, Biochemistry) to humanities (Philosophy, Art History, Creative Writing) and professional programs (Business Administration, Nursing, Architecture)
- **Companies:** 327 major U.S. corporations with associated headquarters cities, covering diverse industries including technology, finance, healthcare, retail, and manufacturing

This vocabulary size enables the generation of approximately $411 \times 461 \times 1{,}002 \times 50 \times 341 \times 101 \times 327 \approx 1.04 \times 10^{17}$ unique person profiles, ensuring minimal memorization pressure and focusing evaluation on semantic understanding rather than rote learning. We pick 1000 samples (6000 entities) in our dataset for the factual knowledge memorization.

### A.3  Question-Answer Templates

Each of the six semantic relations was probed using four distinct question templates, enabling controlled evaluation of template generalization. Table 6 lists all templates used in our experiments.

This template design ensures semantic diversity while maintaining consistent information content, allowing us to test whether the SAE captures abstract semantic relations rather than surface-level linguistic patterns.

| Relation | Train templates (T0–T1) | Held-out templates (T2–T3) |
|---|---|---|
| Birth Date | When was {FULL_NAME} born? 
 On what date was {FULL_NAME} born? | What is {FULL_NAME}'s birth date? 
 Can you tell me the birth date of {FULL_NAME}? |
| Birth City | Where was {FULL_NAME} born? 
 In what city was {FULL_NAME} born? | What is {FULL_NAME}'s birth city? 
 Can you tell me the birth city of {FULL_NAME}? |
| University | Where did {FULL_NAME} go to college? 
 Which college did {FULL_NAME} attend? | What is {FULL_NAME}'s alma mater? 
 Which university did {FULL_NAME} attend? |
| Major | What was {FULL_NAME}'s major? 
 What is {FULL_NAME}'s field of study? | What did {FULL_NAME} study? 
 What field did {FULL_NAME} study in? |
| Employer | Who does {FULL_NAME} work for? 
 What company does {FULL_NAME} work for? | What is {FULL_NAME}'s employer? 
 Which company employs {FULL_NAME}? |
| Work City | Where does {FULL_NAME} work? 
 What city does {FULL_NAME} work in? | What is {FULL_NAME}'s work city? 
 In which city is {FULL_NAME} employed? |

Table 6: Template split for surface-form robustness. T0–T1 are used for training, while T2–T3 are held out for UNSEEN-TEMPLATE evaluation; the split is designed to reduce lexical overlap in relation triggers (*e.g.*, *born* vs. *birth date*, *work for* vs. *employer*).

### A.4 Using an LLM to generate unseen questions

Beyond the fixed template split (T0–T3; Table 6), we explored generating additional *unseen* questions with an external LLM. Concretely, for each 1-hop RC query $(e_1,r)$ (entity $e_1$ and relation $r \in \mathcal{R}$), we few-shot prompt Claude 3.5 Sonnet via the OpenRouter API with two instantiated in-distribution questions (from T0–T1 for the same relation) and ask it to produce exactly two new, *simple and direct* paraphrases that query the same attribute. We enforce basic constraints (use the exact entity name, keep questions short, return exactly two lines); the full prompt is shown in Fig. 10. This procedure produces *instance-specific* unseen questions (unique per $(e_1,r)$), rather than global templates shared across entities.

---

**User prompt used for OOD question generation (verbatim, with placeholders).**

```
You are helping generate simple question variations for a dataset. Your task is to create 2 NEW questions that ask
about {RELATION_DESC} for the person "{FULL_NAME}".

IMPORTANT REQUIREMENTS:
1. Keep questions SIMPLE and DIRECT - similar to these training examples but with SLIGHT variations:
   - "{ID_EXAMPLE_1}"
   - "{ID_EXAMPLE_2}"

2. Make SMALL changes only - change just 1-2 words or reorder slightly
3. Questions must be answerable with a direct factual answer (date, city, university name, etc.)
4. DO NOT ask questions that require explanation or opinion
5. Use the exact name "{FULL_NAME}" in each question
6. Keep questions SHORT (under 15 words)

Generate exactly 2 simple question variations, one per line. Do NOT number them or add any other text.
```

---

Figure 10: Few-shot prompt used to generate OOD paraphrases for each $(p,r)$ pair. {RELATION_DESC} is a short natural-language description of the target relation (*e.g., "the person's birth date"*), and {ID_EXAMPLE_1}–{ID_EXAMPLE_2} are instantiated from the ID training templates (T0–T1).

In practice, many generations drift toward more indirect or complex phrasings that the base GPT-2 backbone cannot reliably answer, even when the questions are semantically well-formed. To isolate the effect of concept

alignment (rather than backbone capacity), we therefore restrict the main experiments to the controlled template split in Table 6 and treat LLM-generated unseen questions as an exploratory stress test.

**Examples.** Compared to the fixed templates, Claude 3.5 Sonnet often introduces indirect phrasing. For BIRTH_DATE: (i) *"Do you happen to know the calendar date that marks Todd Raul Hanson's arrival?"* and (ii) *"I'm trying to find out the calendar date on which Edith Rocky Taylor made her first appearance."* (answer: 12,August,1991). For EMPLOYER: *"I'm curious about the business where Jonathan Kiera Carney earns their living – do you know it?"* (answer: Caterpillar Inc.).

## B  Base Language Model Training

### B.1  Model Architecture

We employed GPT-2 with 124M parameters as the base causal language model, featuring 768-dimensional hidden representations and 12 transformer layers.

### B.2  Training Objective

The model was trained using a two-component curriculum: (1) *biography memorization*, where the model learned to predict entire biography sequences, and (2) *pure question-answering*, where only answer tokens contributed to the loss while question prompts were masked (label = -100).

### B.3  Optimization Hyperparameters

**Selection protocol.** We did not perform an extensive hyperparameter search for base LM training. Unless otherwise noted, we mainly adopt the optimization settings reported in Allen-Zhu & Li (2024) (AdamW with warmup, cosine decay, and gradient clipping) and keep them fixed across all runs. We only ran minimal sanity checks to ensure training stability (no divergence) and to achieve near-saturated train/validation performance on our synthetic curriculum; we did not tune hyperparameters to optimize downstream controllability or binding metrics.

Training was conducted with the following configuration:

- **Maximum training steps:** 80,000
- **Effective batch size:** 96 (distributed across available GPUs)
- **Learning rate schedule:** Linear warmup to $1 \times 10^{-3}$ over 1,000 steps, followed by cosine annealing to $1 \times 10^{-4}$
- **Optimizer:** AdamW with weight decay 0.1 and $\epsilon = 1 \times 10^{-6}$
- **Gradient clipping:** Maximum norm 1.0
- **Maximum sequence length:** 512 tokens
- **Checkpoint frequency:** Every 10,000 steps

### B.4  Activation Collection

Hidden states were extracted from the residual stream at the final token position of the question prompt (immediately before answer generation), representing the point where the model "decides" what information to retrieve. We collected activations from all 12 transformer layers independently to analyze the emergence of semantic binding across network depth.

## C  Concept-Aligned Sparse Autoencoder

### C.1  Model Architecture

Our supervised SAE extends a standard sparse autoencoder with $|\mathcal{R}|$ dedicated relation slots, while delegating residual variance to a large bank of free features. The configuration is:

- **Input dimension:** $d$=768 (GPT-2 hidden size)
- **Total latent features:** $K+|\mathcal{R}|$ with $K \in \{10000, 100000\}$ (task-dependent) and $|\mathcal{R}|$
  - *Free features:* $z_{\text{rest}} \in \mathbb{R}^K$ (unsupervised)
  - *Relation slots:* $z_{\text{concept}} \in \mathbb{R}^{|\mathcal{R}|}$ (supervised; one per relation)
- **Encoder:** $E : \mathbb{R}^d \to \mathbb{R}^{|\mathcal{R}|+K}, \quad z = \text{ReLU}(W_e h + b_e)$ with $z = [z_{\text{concept}}; z_{\text{rest}}]$
- **Decoder:** $D : \mathbb{R}^{|\mathcal{R}|+K} \to \mathbb{R}^d, \quad \hat{h} = W_d z + b_d$

**Binding Mechanism.** We partition $z = [z_{\text{concept}}; z_{\text{rest}}]$ with $z_{\text{concept}} \in \mathbb{R}^{|\mathcal{R}|}$ and define $p(r \mid x) = \text{softmax}(z_{\text{concept}})$. The binding loss applies cross-entropy with the one-hot gold relation, encouraging the correct relation slot to dominate.

## C.2 Multi-Stage Training Protocol

Training proceeds in two stages to ensure stable convergence:

- **Stage 1 (Reconstruction-Only):** 100 epochs focusing solely on autoencoding quality before introducing binding constraints
- **Stage 2 (Full Supervision):** 500 epochs with complete loss function

## C.3 Loss Function

The total training objective (see Section 4 of the main paper) combines six components. In Stage 1 (100 epochs), only reconstruction loss is active to stabilize the encoder-decoder. In Stage 2 (500 epochs), all six losses are jointly optimized:

$$\mathcal{L}_{\text{total}} = \lambda_{\text{recon}}\mathcal{L}_{\text{recon}} + \lambda_{\text{sparse}}\mathcal{L}_{\text{sparse}} + \lambda_{\text{align}}\mathcal{L}_{\text{align}} \\ + \lambda_{\text{ortho}}\mathcal{L}_{\text{ortho}} + \lambda_{\text{value}}\mathcal{L}_{\text{value}} \tag{2}$$

Each component serves a specific purpose:

**Reconstruction Loss.**

$$\mathcal{L}_{\text{recon}} = \text{MSE}(\hat{h}, h) = \frac{1}{d}\sum_{i=1}^{d}(\hat{h}_i - h_i)^2 \tag{3}$$

Measures mean squared error between original activation $h$ and reconstructed activation $\hat{h} = W_{\text{dec}} \cdot z$, where $d = 768$ is the hidden dimension. This ensures the SAE preserves information necessary for the language model's downstream predictions while learning a compressed latent representation.

**Sparsity Loss.**

$$\mathcal{L}_{\text{sparse}} = \frac{1}{B \cdot n_{\text{free}}}\sum_{b=1}^{B}\sum_{j=1}^{n_{\text{free}}}|z_{b,j}| \tag{4}$$

Enforces L1 penalty on the large number of free slots across batch size $B$, encouraging the model to activate only a small subset of features per sample. Sparse activations improve interpretability by ensuring each latent feature captures distinct semantic properties rather than distributing information diffusely.

**Alignment Loss.**

$$\mathcal{L}_{\text{align}} = \text{CrossEntropy}(\text{softmax}(z_{\text{rel}}), y) \tag{5}$$

$$= -\frac{1}{B}\sum_{b=1}^{B}\sum_{r=1}^{6} y_{b,r} \log \frac{\exp(z_{b,n_{\text{free}}+r})}{\sum_{r'=1}^{6}\exp(z_{b,n_{\text{free}}+r'})}. \tag{6}$$

Provides supervised guidance where $y$ is a one-hot vector with $y_{b,\text{rule\_idx}_b} = 1$ indicating the ground-truth relation type. This cross-entropy loss over softmax-normalized relation slot activations enforces that the 6 relation slots (indices $n_{\text{free}} + 1$ through $n_{\text{free}} + 6$) form a probability distribution with mass concentrated on the correct semantic relation. This is the binding loss used in all main experiments, enabling explicit classification of question types to specific latent dimensions.

**Orthogonality Loss.**

$$\mathcal{L}_{\text{ortho}} = \sum_{r=1}^{6} \sum_{j=1}^{n_{\text{free}}} \left( \frac{1}{B} \sum_{b=1}^{B} (z_{b,n_{\text{free}}+r} - \bar{z}_r)(z_{b,j} - \bar{z}_j) \right)^2 \tag{7}$$

Enforces statistical independence between supervised relation slots and unsupervised free slots by minimizing their cross-covariance. This prevents relation slots from encoding information already captured by free features, ensuring clean separation between task-specific and general-purpose representations.

**Value Prediction Loss.** To encourage each relation slot to be task-informative, *i.e.*, to carry evidence that supports answer prediction beyond merely signaling its identity, we add an auxiliary value objective. For each training example $b$, we take the activation of the gold relation slot $r_b = \text{rule\_idx}_b$ and predict the first token of the answer via a relation-specific head:

$$\mathcal{L}_{\text{value}} = \frac{1}{B} \sum_{b=1}^{B} \text{CrossEntropy}\Big( V_{r_b}\big(z_{b,n_{\text{free}}+r_b}\big), t_b \Big), \tag{8}$$

where $t_b$ is the first token of the ground-truth answer and $V_r$ is a two-layer MLP mapping the scalar slot activation to vocabulary logits. We backpropagate $\mathcal{L}_{\text{value}}$ jointly with the reconstruction and binding losses so that concept slots remain predictive of the target output, while the large free-feature bank captures residual variation needed for high-fidelity reconstruction. The value heads serve as an auxiliary training signal (and diagnostic readout) and are not used in inference-time interventions.

**Independence validation (not trained).** Although we do not optimize an explicit free-slot independence term, we validate redundancy via an *independence score* computed as the squared off-diagonal covariance among free features:

$$\mathcal{L}_{\text{indep}} = \sum_{i \neq j} \left( \frac{1}{B} \sum_{b=1}^{B} (z_{b,i} - \bar{z}_i)(z_{b,j} - \bar{z}_j) \right)^2, \tag{9}$$

where $\bar{z}_i = \frac{1}{B} \sum_{b=1}^{B} z_{b,i}$. Without Stage 1, this score explodes ($4.76 \rightarrow 157.80$, $+3213.7\%$), indicating highly correlated and redundant free features.

## C.4 Loss Weight Selection

**Selection protocol.** We did not perform an extensive hyperparameter search. Unless explicitly noted, we use **standard/default SAE choices** and keep them fixed across all runs. We only ran minimal sanity checks to avoid degenerate behavior (*e.g.*, dead features or collapse), and we did not tune weights to maximize reported metrics.

The loss components are weighted to balance competing objectives:

- $\lambda_{\text{recon}} = 1.0$ — High priority is given to faithful reconstruction to maintain model performance. This ensures the SAE does not distort the information flow through the network.
- $\lambda_{\text{sparse}} = 1 \times 10^{-3}$ — Gentle L1 penalty on free slots. This small weight prevents over-suppression of activations while still encouraging selective feature usage. Stronger sparsity penalties ($\lambda > 10^{-2}$) caused excessive dead neurons and degraded reconstruction quality in preliminary experiments.
- $\lambda_{\text{align}} = 1.0$ — Strong supervision signal to ensure reliable slot-relation binding. This weight is balanced with reconstruction to achieve $> 95\%$ binding accuracy on in-distribution templates in Fig. 4.

- $\lambda_{\text{ortho}} = 1 \times 10^{-2}$ — Moderate orthogonality constraint between relation and free slots. This maintains separation between supervised and unsupervised features, preventing information leakage that could compromise the interpretability of relation slots.
- $\lambda_{\text{value}} = 0.5$ — Balanced weight for answer prediction. This auxiliary task provides a training signal to ensure relation slots encode semantically meaningful information, but is weighted lower than alignment to avoid dominating the optimization.

### C.5 Training Hyperparameters

**Selection protocol.** We use standard/default optimizer settings (AdamW) and do not tune hyperparameters for peak performance; we only ran minimal sanity checks for training stability.

**Optimizer configuration.** We use AdamW with the following settings (Loshchilov & Hutter, 2019):

- **Learning rate:** $1 \times 10^{-3}$ (constant, no warmup or decay)
- **Weight decay:** $0.0$ (L2 regularization disabled to avoid interfering with explicit sparsity constraints)
- **Betas:** $(0.9, 0.999)$ (default momentum coefficients)
- **Epsilon:** $1 \times 10^{-8}$ (numerical stability constant)
- **Batch size:** 64 samples per update
- **Gradient clipping:** None (training was stable without clipping)

**Training Schedule.** The constant learning rate without decay was chosen because the two-stage training protocol naturally provides curriculum learning: Stage 1 establishes a good initialization for the encoder-decoder using traditional SAE framework Shu et al. (2025), after which Stage 2 refines the latent structure, as defined as SAE post-training. Preliminary experiments with cosine annealing showed no improvement over constant learning rate for this setting.

## D Evaluation Metrics

This section provides detailed mathematical definitions for all evaluation metrics reported in Section 5 of the main paper.

### D.1 Binding Accuracy Metrics

We evaluate the quality of semantic binding using multiple complementary metrics reported in Table 1 and Fig. 3 of the main paper:

**Slot Binding Accuracy.** The fraction of questions that activate the correct relation slot, defined as:

$$\text{Acc}_{\text{binding}} = \frac{1}{N} \sum_{i=1}^{N} \mathbf{1} \left[ \underset{j}{\arg\max} \ z_{\text{rel},j}^{(i)} = r_i \right] \tag{10}$$

where $z_{\text{rel}}^{(i)}$ is the relation slot activation vector for question $i$ and $r_i$ is the ground-truth relation. This metric measures one-to-one mapping quality and is the primary metric reported in Table 1.

**Top-$k$ Accuracy.** A relaxed metric checking whether the true relation slot appears in the top-$k$ predictions:

$$\text{Acc}_{\text{top-}k} = \frac{1}{N} \sum_{i=1}^{N} \mathbb{1}[r_i \in \text{TopK}(z_{\text{rel}}^{(i)})] \tag{11}$$

This metric is useful for understanding near-miss cases where the correct slot has high but not maximal activation.

**Margin.** The logit difference between the top-1 and top-2 slot predictions, measuring binding confidence:

$$\text{Margin} = \frac{1}{N} \sum_{i=1}^{N} (z_{\text{rel},j_1}^{(i)} - z_{\text{rel},j_2}^{(i)}) \tag{12}$$

where $j_1$ and $j_2$ are the indices of the highest and second-highest activations. Higher margins indicate more confident and unambiguous binding. We report average margins in Section 5.3 when analyzing binding robustness across layers.

**Answer Accuracy.** Exact-match accuracy for generated answers:

$$\text{Acc}_{\text{answer}} = \frac{1}{N} \sum_{i=1}^{N} \not\Vdash[\hat{a}_i = a_i] \tag{13}$$

where the normalization function handles multiple date formats (*e.g.*, *"Day, Month, Year"*; *"Month Day, Year"*; *"YYYY-MM-DD"*) to avoid penalizing formatting differences.

**Swap Intervention Accuracy (Causal Control).** Measures the full language model's answer generation after latent manipulation in swap experiments (Section 5.4). After modifying relation slot activations ($z_i^{\text{orig}} \leftarrow 0$, $z_j^{\text{target}} \leftarrow \alpha$), we decode to obtain $\hat{h} = W_{\text{dec}} \cdot z'$ and feed this modified activation through the remaining transformer layers to generate text using the LLM's standard autoregressive generation. The value heads are *not used* in these experiments. This validates causal control over model behavior. Optimal swap success: 85% at $\alpha \approx 2$ (Layer 6).

The swap intervention accuracy validates that these representations causally influence the full model's behavior.

**Diagonal Accuracy.** Quantifies the one-to-one mapping quality between ground-truth relations and predicted slots using the confusion matrix:

$$\text{Diag} = \frac{1}{R} \sum_{i=1}^{R} C_{ii} \tag{14}$$

where $R = 6$ is the number of relations and $C$ is the normalized confusion matrix with $C_{ij} = \frac{\#\{r=i, \hat{r}=j\}}{\#\{r=i\}}$. Perfect binding yields Diag $= 1.0$, while random assignment gives Diag $\approx 0.167$. Confusion matrices are visualized in Fig. 3 for layer-wise analysis.

### D.2 Reconstruction Quality

We measure the fidelity of the autoencoder's reconstruction using mean squared error:

$$\text{MSE}_{\text{recon}} = \frac{1}{N} \sum_{i=1}^{N} \|h^{(i)} - \hat{h}^{(i)}\|^2 \tag{15}$$

where $h^{(i)}$ is the original 768-dimensional activation vector from GPT-2's residual stream and $\hat{h}^{(i)} = W_{\text{dec}} \cdot z^{(i)}$ is the reconstructed activation after encoding and decoding through the SAE. The reconstruction target is the raw activation, not normalized or preprocessed. Layer 6 achieves MSE $\approx 7.42 \times 10^{-2}$, representing a trade-off between reconstruction fidelity and semantic structure—early layers achieve lower MSE (*e.g.*, Layer 0: $6.53 \times 10^{-5}$) but lack meaningful concept binding. The relatively higher MSE in middle layers reflects the cost of enforcing interpretable slot structure while maintaining sufficient information for downstream task performance.

### D.3 Swap Controllability

To test whether ontology-aligned slots serve as usable *control knobs*, we measure the success rate of answer swaps under inference-time steering along decoded SAE directions (protocol in §4.4; implementation details in §B.4).

**Intervention.** Let $h_{\ell,t} \in \mathbb{R}^d$ denote the residual stream at layer $\ell$ and token position $t$, and let $W_{\mathrm{dec}} \in \mathbb{R}^{d \times K}$ be the SAE decoder. For a target concept slot $j \in \{1, \ldots, K\}$, define the decoded direction $v_j := W_{\mathrm{dec}} e_j$. Given a question whose gold relation is $r^\star$ and a chosen target slot $j \neq \pi(r^\star)$, we steer the residual stream by

$$h'_{\ell,t} \;=\; h_{\ell,t} + \alpha v_j \;=\; h_{\ell,t} + W_{\mathrm{dec}}(\alpha e_j), \tag{16}$$

then continue the forward pass through the remaining transformer layers and decode the answer. We sweep amplification strengths

$$\alpha \in \{0.1, 0.5, 1, 2, 5, 10, 20, 50, 100, 200, 500, 1000\}. \tag{17}$$

**Swap success.** For swap trial $m$, let $p_m$ be the subject entity and let $r_m^{\mathrm{tgt}}$ be the target relation corresponding to the amplified slot $j$. Let $\hat{a}_m^{\mathrm{swap}}$ be the generated answer under the intervention. We count the swap as successful if the model outputs the gold value for the *target* relation, $g(p_m, r_m^{\mathrm{tgt}})$:

$$\mathrm{Swap}_\alpha = \frac{1}{M} \sum_{m=1}^{M} \not\Vdash\big[\hat{a}_m^{\mathrm{swap}} = g(p_m, r_m^{\mathrm{tgt}})\big]. \tag{18}$$

Full metric definitions are provided in §D.

As shown in Figure 6, controllability is maximized at moderate amplification ($\alpha \approx 2$): at Layer 6, swap success rises from 0.54 at $\alpha = 1$ to 0.85 at $\alpha = 2$, but drops to 0.23 at $\alpha = 10$, indicating that over-amplification destabilizes the intervention.

## E  Layer-wise SAE Feature Comparison

This section presents a comprehensive comparison of top-50 activated features across all 12 transformer layers (Layer 0–11) of GPT-2 (Fig. 11). Each visualization compares two conditions: (1) **with SAE post-training** (supervised sparse autoencoder applied), and (2) **without SAE post-training** (baseline activations). This comparison reveals how supervised alignment shapes feature representations and demonstrates the emergence of clean semantic binding in middle layers, while also highlighting artifacts that appear in deeper layers without SAE regularization.

### E.1  Analysis

These layer-wise visualizations reveal several critical insights about the role of supervised SAE training across network depth:

**Early Layers (0–4): Limited Semantic Structure.** In shallow layers, both conditions (with and without SAE) show relatively diffuse activation patterns with weak relation-specific structure. This reflects that early transformer layers primarily process local token-level features and have not yet formed abstract semantic representations suitable for clean concept binding. The SAE provides marginal improvements but cannot overcome the fundamental limitation that these layers lack the representational capacity for high-level semantic concepts.

**Middle Layers (5–8): Emergence of Clean Binding.** The most dramatic differences appear in middle layers, particularly Layer 6. With SAE post-training, we observe sharp, diagonal activation patterns indicating successful one-to-one binding between ontological relations and designated slots. Without SAE supervision, these same layers show more scattered, overlapping activations that fail to achieve clean separation between semantic concepts. This demonstrates that while middle layers contain the raw representational power for concept binding, explicit supervision through the SAE's multi-objective loss is essential to crystallize these latent capabilities into interpretable, controllable structure.

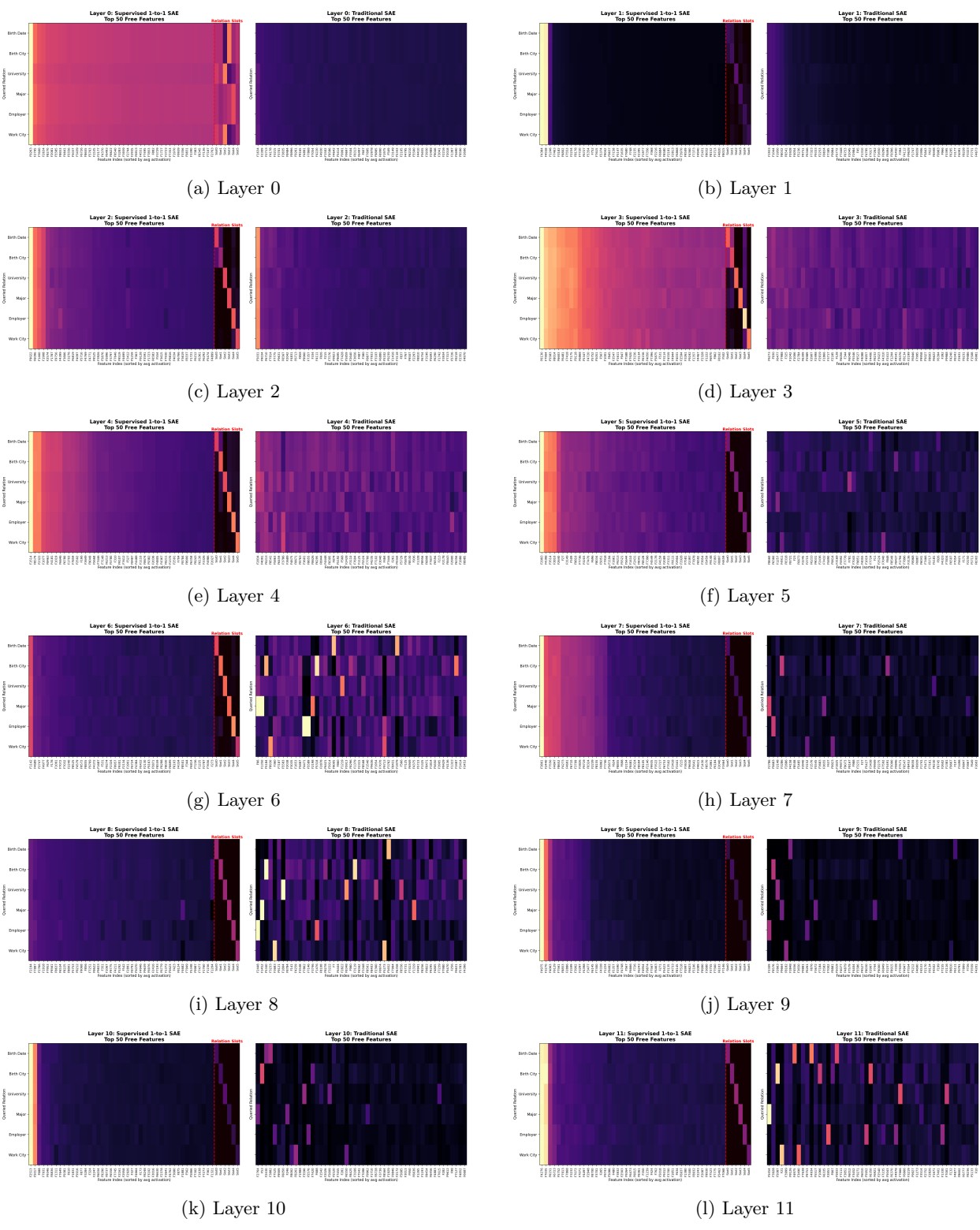

Figure 11: Layer-wise top-50 feature activations in GPT-2 with and without SAE post-training. Early layers (0–3) remain diffuse and entangled. Middle layers (4–8) show maximum SAE benefit, with Layer 6 achieving optimal semantic binding and diagonal accuracy. Deep layers (9–11) without SAEs exhibit irregular patterns and over-compression, whereas SAE-trained models maintain structured representations.

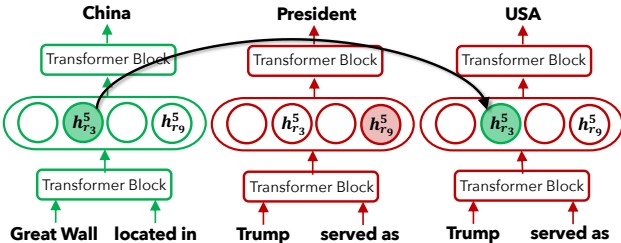

Figure 12: Illustrative schematic of our swap intervention (Orig relation → Swap relation).

**Deep Layers (9–11): Artifacts Without SAE.** A particularly interesting phenomenon emerges in deeper layers without SAE post-training. Starting around Layer 6 and becoming more pronounced in Layers 9–11, the baseline (no SAE) condition exhibits strange, irregular activation patterns—potentially including sparse, extreme activations, feature collapse, or unexpected clustering. These artifacts likely reflect the model's aggressive compression of task-relevant information in preparation for final output generation. The supervised SAE mitigates these irregularities by enforcing reconstruction fidelity, sparsity constraints, and orthogonality between relation and free slots, resulting in more stable and interpretable features even in deep layers.

**Optimal Layer for Intervention.** These visualizations provide empirical justification for choosing Layer 6 as the primary layer for semantic intervention experiments (as reported in Section 5 of the main paper). Layer 6 achieves: (1) mature semantic representations that support clean binding, (2) strong separation between concepts under SAE training, (3) minimal artifacts compared to deeper layers, and (4) acceptable reconstruction error that preserves model functionality.

**The Role of Supervised Training.** Across all layers, SAE post-training consistently produces more structured, interpretable activation patterns. The supervised losses—particularly alignment loss (enforcing relation-slot correspondence) and orthogonality loss (separating supervised and unsupervised features)—act as powerful inductive biases that shape the latent space into a form amenable to human interpretation and causal intervention. Without this supervision, even layers with rich semantic content fail to expose that structure in an accessible format.

## F Additional Qualitative Swap Examples

We provide qualitative swap examples to complement the quantitative controllability results (§D.3, §5.3). Unless otherwise noted, all examples use Layer 6 and intervention strength $\alpha=2$. In each example, we start from a question whose gold relation is the *Orig* type and intervene by amplifying the decoded direction of a different concept slot (*Swap*), aiming to make the model answer with the corresponding target attribute.

### F.1 Swapping Schematic

Fig. 12 illustrates the swap intervention: we steer the layer-$\ell$ residual representation along the decoded direction of a target (Swap) concept slot to flip the predicted answer type from the original relation.

### F.2 Correct Swap Examples

Table 7 reports successful swaps. **Orig→Swap** denotes the original relation queried and the target relation we force via intervention. **Target** is the gold value $g(p,\text{Swap})$ for the same person $p$, and **Generated** is the model output after the swap (counted as correct when it matches the Target).

The swap error analysis, including per-concept category retention statistics and a representative failure case, is presented in §5.4 of the main paper.

| Orig → Swap | Question | Target | Generated |
|---|---|---|---|
| COMPANY_CITY → UNIVERSITY | What is Grace Wendy Rivera's work city? | Florida International University | Florida International University |
| COMPANY_CITY → MAJOR | What is Grace Wendy Rivera's work city? | Electrical Engineering | Electrical Engineering |
| COMPANY_CITY → EMPLOYER | What is Grace Wendy Rivera's work city? | Blackstone | Blackstone |
| UNIVERSITY → BIRTH_DATE | Where did Thomas Heath Stafford go to college? | 2, March, 1981 | 2, March, 1981 |
| UNIVERSITY → MAJOR | Where did Thomas Heath Stafford go to college? | Dance | Dance |
| BIRTH_DATE → WORK_CITY | When was Megan Kian Valencia born? | Framingham, MA | Framingham, MA |
| BIRTH_CITY → BIRTH_DATE | Where was Angela Maddox Gates born? | 27, November, 1950 | 27, November, 1950 |
| MAJOR → UNIVERSITY | What was Jennifer Donovan Pruitt's major? | University of Wisconsin–Madison | University of Wisconsin–Madison |

Table 7: Correct swap examples from the evaluation set (Layer 6; $\alpha$=2).

## G Mutual Information Gap Analysis

To quantify disentanglement beyond fragmentation and concentration, we compute the Mutual Information Gap (MIG) (Chen et al., 2018) between supervised slot activations and concept labels. For each concept $k$, we identify the two slots with highest mutual information $I(z_j; k)$ and compute

$$\text{MIG}_k = \frac{I(z_{j_1}; k) - I(z_{j_2}; k)}{H(k)}, \tag{19}$$

where $H(k)$ is the entropy of concept $k$. Higher MIG indicates that each concept is captured by a single dominant slot.

| Setting | AlignSAE | Standard SAE | Ratio |
|---|---|---|---|
| 1-Hop (6 concepts) | **0.173** | 0.000 | $\infty$ |
| 2-Hop (20 relations) | **0.035** | 0.004 | $9.2\times$ |

Table 8: Mutual Information Gap (MIG). Standard SAE uses identical architecture trained with only reconstruction and sparsity losses. An MIG of 0.000 is expected for the standard SAE: without concept supervision, no single unsupervised feature has any incentive to exclusively encode a specific concept—information is distributed across many features by design.

Table 8 reports the results: AlignSAE achieves consistent per-concept MIG values ranging from 0.146 to 0.192 across the 6 concepts (1-hop) and 0.022 to 0.039 across the 20 relations (2-hop). The lower absolute 2-hop MIG reflects the harder 20-way discrimination, but the $9.2\times$ improvement over the standard SAE demonstrates substantial disentanglement gains.

## H Latent Space Visualization (t-SNE)

Figure 13 visualizes the latent space before and after alignment via t-SNE projections. For 1-hop, raw hidden activations (768-dim) show partial but overlapping clustering by concept; AlignSAE supervised slot activations (6-dim) produce tight, fully separated clusters for all 6 concepts. For 2-hop, raw activations form a largely undifferentiated cloud; AlignSAE slots create 20 distinct clusters. Figure 14 shows the grokking progression: t-SNE of supervised slot activations at training epochs 5, 10, 25, 50, and 100. At epoch 5, no structure is visible. By epoch 25, clusters begin forming. By epoch 100, clean one-to-one separation emerges.

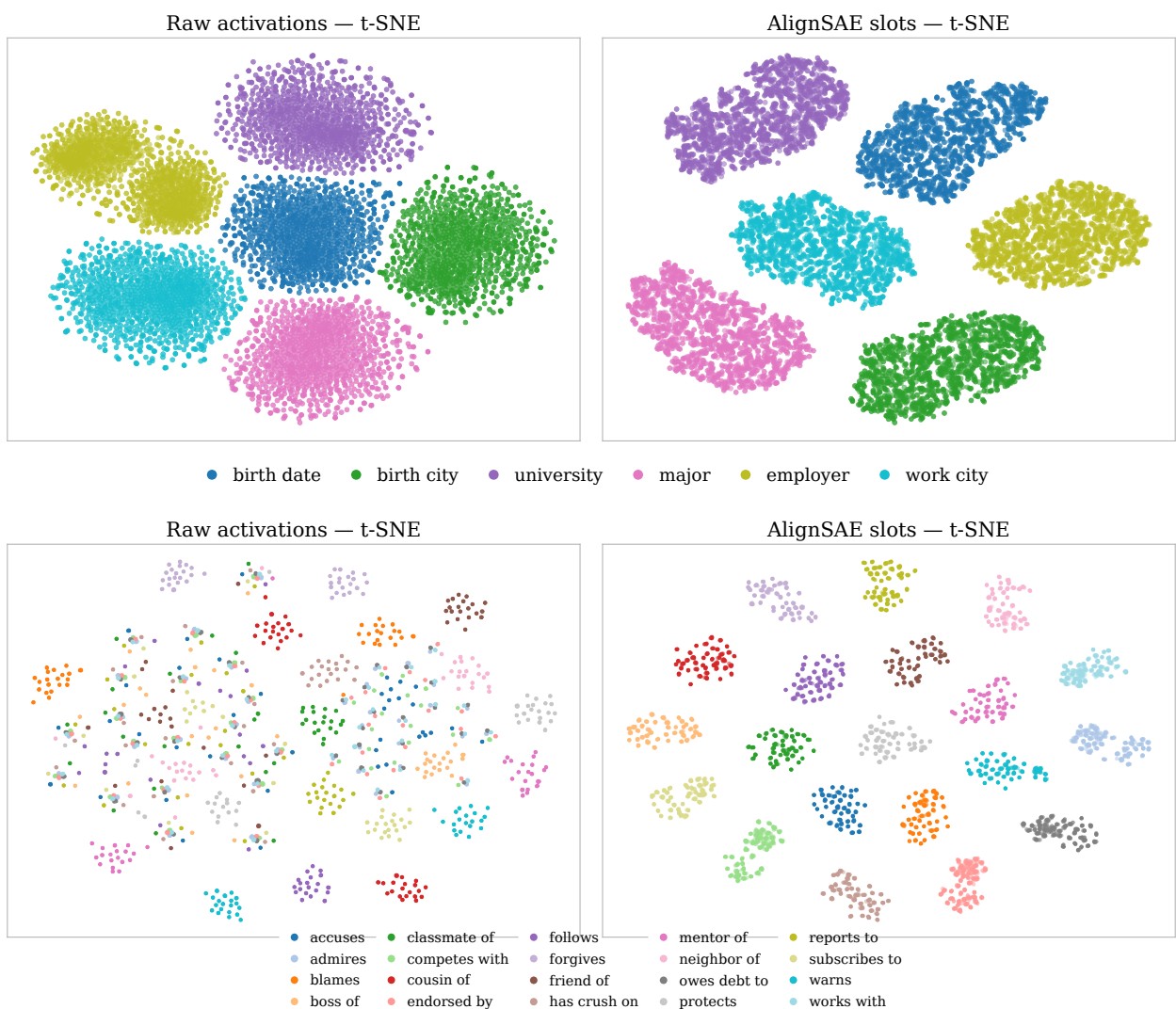

Figure 13: t-SNE projections of raw activations (left) vs. AlignSAE supervised slot activations (right), colored by concept label. Top: 1-hop (6 concepts). Bottom: 2-hop (20 relations).

## I  Steering Side-Effects Analysis

A practical steering interface must not only induce the target concept but also preserve other model capabilities. We evaluate side-effects via perplexity analysis and free-form generation quality.

**Perplexity vs. Amplification.**  We compute perplexity on held-out QA data after intervening at different $\alpha$ values, comparing correct-slot steering (boosting the true concept) vs. wrong-slot steering (boosting an incorrect concept). As shown in Fig. 15, correct-slot steering reduces perplexity by over 5 orders of magnitude from the no-SAE baseline ($\sim$3,438 at $\alpha$=50 for 1-hop), then plateaus. Wrong-slot steering maintains high perplexity across all $\alpha$ values. This asymmetry confirms that AlignSAE steering provides a precise, targeted signal: it helps when aligned with the ground truth and hurts when misaligned. The SAE reconstruction baseline (encode→decode with no slot modification) already improves perplexity over the no-SAE condition.

**Free-Form Generation Quality.**  We evaluate generation from neutral prompts ("Once upon a time,", "The weather today is", etc.) with and without steering. Since the model was fine-tuned on a predefined

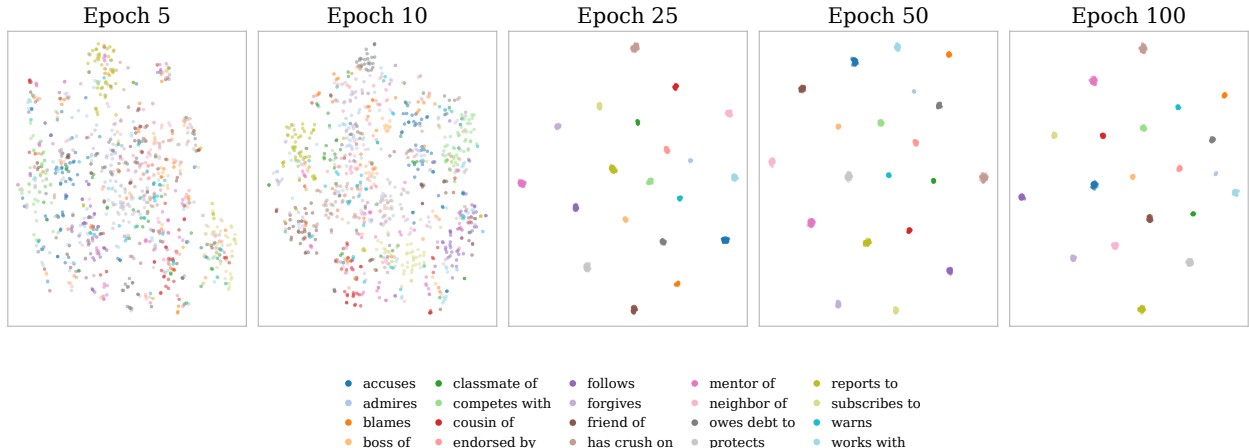

Figure 14: Grokking progression: t-SNE of 2-hop supervised slot activations at epochs 5, 10, 25, 50, 100.

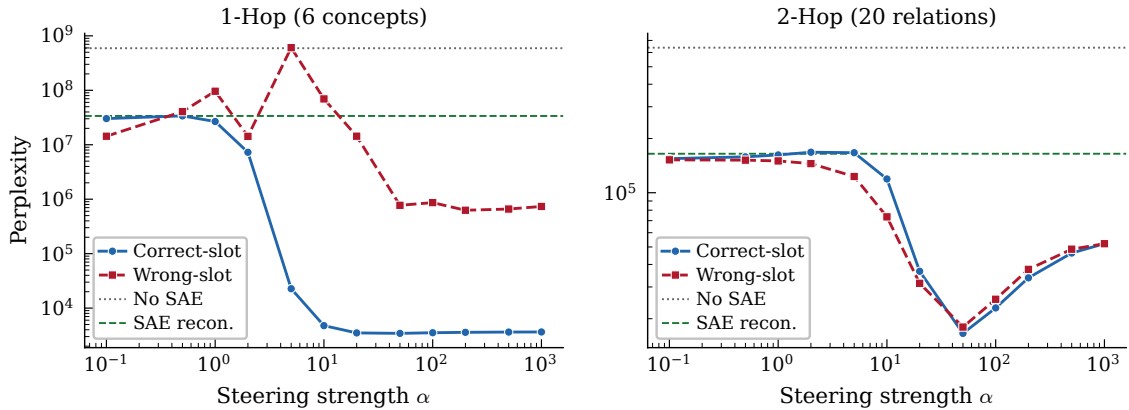

Figure 15: Perplexity vs. steering strength $\alpha$. Correct-slot steering (blue) dramatically reduces perplexity; wrong-slot steering (red) does not. Dashed lines show no-SAE and SAE-reconstruct baselines.

ontology of structured biographical paragraphs and QA pairs, it is *designed* to produce ontology-grounded factual outputs rather than free-form creative text.

Table 9 shows representative outputs. Without steering, the model produces generic biographical text. When we steer toward a specific concept (birth_date, major, or work_city), the *first generated token* immediately reflects the target concept—a date, a field of study, or a city name—while the continuation remains fluent biographical text. The model's domain-specific behavior is preserved; only the *leading concept* changes. This first-token effect demonstrates that AlignSAE's 1-to-1 concept binding provides a precise causal handle: the steered slot directly influences the model's next-token distribution at generation onset.

An LLM judge (Claude Haiku 4.5) rated 70 generations across fluency, coherence, and concept steering success (1–5 scale). Table 10 reports the full results. Key findings: (i) steering preserves fluency (baseline ≈ steered: 2.0 vs. 2.1 for 1-hop, 4.0 vs. 3.9 for 2-hop); (ii) concept steering success is 3.9/5 for 1-hop; (iii) results are consistent across $\alpha$=10 and $\alpha$=100. Because the concepts are well isolated via AlignSAE's 1-to-1 binding, steering has *fewer undesirable side-effects*—it changes *what* the model generates without degrading *how well* it generates.

Since the models are fine-tuned exclusively on synthetic data, general-purpose benchmarks (e.g., HellaSwag) would not yield meaningful signal. We adopt a controlled benchmark within the model's own data distribution: perplexity on held-out QA, swap success rates, and LLM-as-Judge assessment measure whether steering disrupts the model's learned capabilities on its training task.

| Condition | First tokens generated |
|---|---|
| No steering | *Whitman College counts She among its graduates. She works in Purchase, NY. Her field of study was Nursing...* |
| Steer → birth_date ($\alpha$=10) | **27,July,1991**, *Bryan Caleb Atkins was born. He is professionally active in Fort Worth, TX...* |
| Steer → major ($\alpha$=10) | **French**. *According to records, Maurice Hudson Tran was born on 28,September,1953...* |
| Steer → work_city ($\alpha$=10) | **Dallas, TX**. *He earned a degree in Nursing. He is a key part of the AT&T team...* |

Table 9: Free-form generation examples (1-hop, $\alpha$=10). Bold text marks the first tokens generated, which immediately reflect the steered concept. Continuations remain fluent biographical text.

| Setting | Condition | Fluency | Coherence | Steering | $n$ |
|---|---|---|---|---|---|
| | Baseline | 2.0 | 1.0 | — | 5 |
| 1-Hop | Steered $\alpha$=10 | 2.1 | 1.1 | 3.9 | 15 |
| | Steered $\alpha$=100 | 2.0 | 1.1 | 3.9 | 15 |
| | Baseline | 4.0 | 1.2 | — | 5 |
| 2-Hop | Steered $\alpha$=10 | 4.0 | 1.3 | 2.5 | 15 |
| | Steered $\alpha$=100 | 3.7 | 1.3 | 2.4 | 15 |

Table 10: LLM-as-Judge evaluation of free-form generation quality (1–5 scale). Steering preserves fluency and coherence while successfully injecting the target concept in 1-hop.

## J   Scalability: FB15K with 91 Relations

To assess scalability beyond small relation sets, we evaluate ALIGNSAE on a subset of FB15K (Bordes et al., 2013) with 91 relations. We retain the same SAE architecture and training protocol, increasing only the supervised slot count to $|\mathcal{R}|$=91. All experiments use Layer 6.

**Dataset Construction.**   FB15K contains 91 relation types. For each relation we sample up to 90 unique head entities; relations with fewer than 90 heads retain their original count (see Table 12 for per-relation statistics). QA and biography templates are generated with GPT following the same procedure as the 1-hop setting.

**Binding and Reconstruction.**   Table 11 summarizes the binding and reconstruction results. Train binding accuracy reaches 98.7%, and the confusion matrix diagonal accuracy is 95.6% (Fig. 16), indicating that ALIGNSAE successfully scales to 91 supervised slots with near-perfect in-distribution binding. Out-of-distribution binding (unseen templates/persons) is 75.7%, substantially above the $1/91 \approx 1.1\%$ chance level. Reconstruction MSE remains comparable to the smaller settings. The two relations with noticeably lower binding accuracy—*/base/locations/continents/countries_within* (4 entities) and */broadcast/content/artist* (6 entities)—have very few training examples, explaining their weaker performance.

| Metric | Value |
|---|---|
| Train binding accuracy (slot top-1) | 0.987 |
| Test binding accuracy (slot top-1, OOD) | 0.757 |
| Reconstruction MSE | 0.083 |
| Confusion matrix diagonal accuracy | 0.956 |

Table 11: ALIGNSAE performance in the 91-relation FB15K setting.

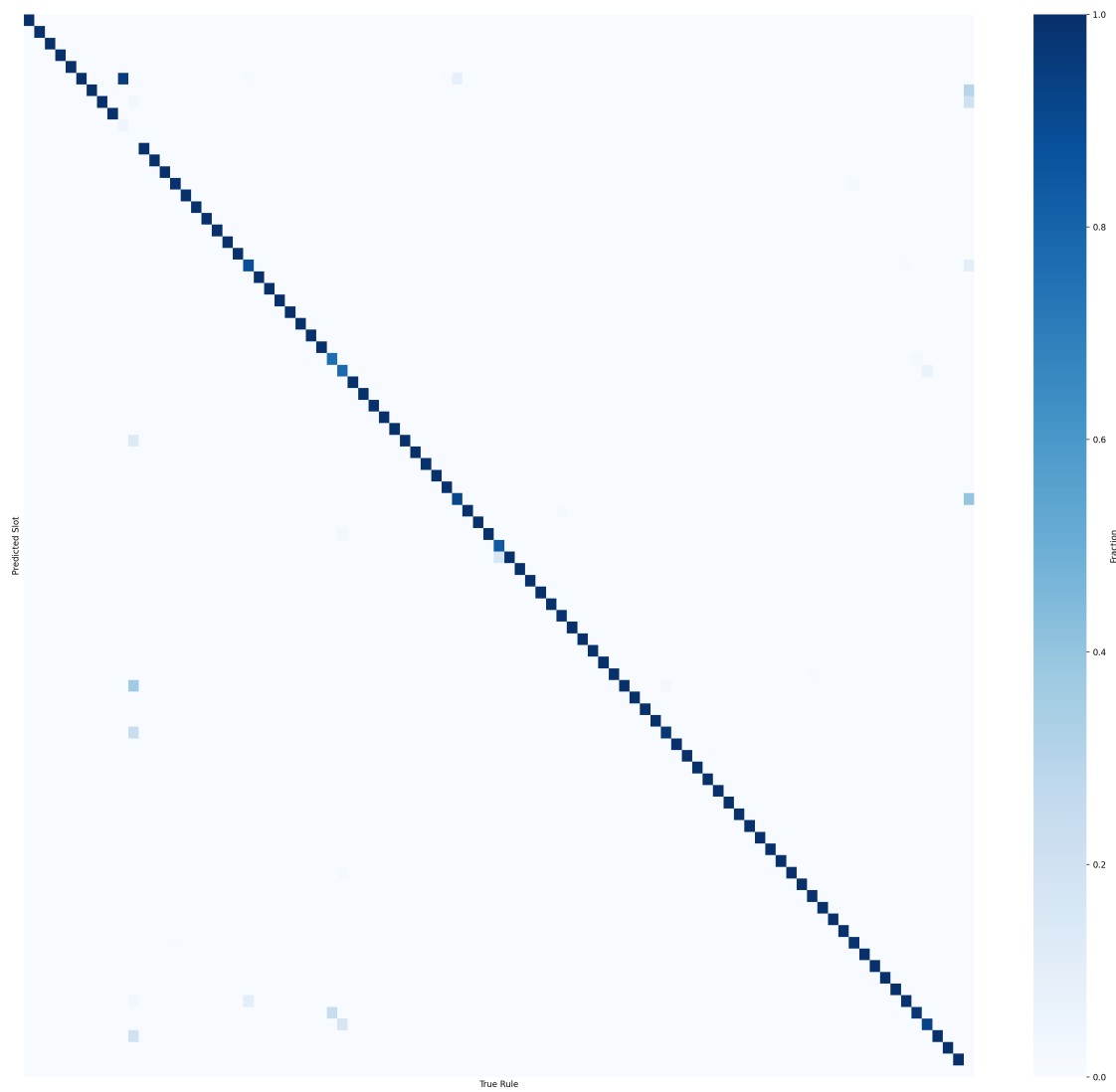

Figure 16: Normalized confusion matrix for 91-relation slot binding at Layer 6. The strong diagonal confirms that ALIGNSAE maintains near-perfect concept–slot alignment even at this scale.

| Idx | Relation | #Ent | Idx | Relation | #Ent |
|---|---|---|---|---|---|
| 0 | /award/award_category/category_of | 90 | 45 | /location/country/second_level_divisions | 11 |
| 1 | /award/award_category/disciplines_or_subjects | 90 | 46 | /location/hud_county_place/county | 90 |
| 2 | /base/aareas/schema/administrative_area/administrative_area_type | 90 | 47 | /location/hud_county_place/place | 90 |
| 3 | /base/aareas/schema/administrative_area/administrative_parent | 90 | 48 | /location/location/contains | 90 |
| 4 | /base/americancomedy/celebrity_impressionist/celebrities_impersonated | 8 | 49 | /location/location/partially_contains | 90 |
| 5 | /base/biblioness/bibs_location/country | 90 | 50 | /location/location/time_zones | 90 |
| 6 | /base/biblioness/bibs_location/state | 90 | 51 | /location/us_county/county_seat | 90 |
| 7 | /base/culturalevent/event/entity_involved | 65 | 52 | /media_common/netflix_genre/titles | 90 |
| 8 | /base/eating/practicer_of_diet/diet | 90 | 53 | /medicine/disease/notable_people_with_this_condition | 22 |
| 9 | /base/locations/continents/countries_within | 4 | 54 | /medicine/disease/risk_factors | 60 |
| 10 | /broadcast/content/artist | 6 | 55 | /medicine/symptom/symptom_of | 23 |
| 11 | /business/business_operation/industry | 90 | 56 | /music/artist/origin | 90 |
| 12 | /dataworld/gardening_hint/split_to | 90 | 57 | /music/genre/artists | 90 |
| 13 | /education/educational_institution/campuses | 90 | 58 | /music/genre/parent_genre | 90 |
| 14 | /education/educational_institution/colors | 90 | 59 | /music/instrument/family | 81 |
| 15 | /education/educational_institution/school_type | 90 | 60 | /music/instrument/instrumentalists | 86 |
| 16 | /education/educational_institution_campus/educational_institution | 90 | 61 | /music/record_label/artist | 90 |
| 17 | /education/university/fraternities_and_sororities | 90 | 62 | /olympics/olympic_games/participating_countries | 41 |
| 18 | /film/director/film | 90 | 63 | /olympics/olympic_games/sports | 43 |
| 19 | /film/film/cinematography | 90 | 64 | /organization/organization/place_founded | 90 |
| 20 | /film/film/costume_design_by | 90 | 65 | /organization/organization_founder/organizations_founded | 90 |
| 21 | /film/film/country | 90 | 66 | /people/cause_of_death/people | 66 |
| 22 | /film/film/edited_by | 90 | 67 | /people/deceased_person/place_of_burial | 90 |
| 23 | /film/film/executive_produced_by | 90 | 68 | /people/deceased_person/place_of_death | 90 |
| 24 | /film/film/featured_film_locations | 90 | 69 | /people/ethnicity/geographic_distribution | 40 |
| 25 | /film/film/film_art_direction_by | 86 | 70 | /people/ethnicity/languages_spoken | 70 |
| 26 | /film/film/film_festivals | 90 | 71 | /people/ethnicity/people | 75 |
| 27 | /film/film/film_format | 90 | 72 | /people/person/gender | 90 |
| 28 | /film/film/film_production_design_by | 90 | 73 | /people/person/languages | 90 |
| 29 | /film/film/genre | 90 | 74 | /people/person/nationality | 90 |
| 30 | /film/film/language | 90 | 75 | /people/person/place_of_birth | 90 |
| 31 | /film/film/music | 90 | 76 | /people/person/profession | 90 |
| 32 | /film/film/prequel | 90 | 77 | /people/person/religion | 90 |
| 33 | /film/film/produced_by | 90 | 78 | /people/profession/specialization_of | 90 |
| 34 | /film/film/production_companies | 90 | 79 | /sports/sports_team/colors | 90 |
| 35 | /film/film/story_by | 90 | 80 | /sports/sports_team/sport | 90 |
| 36 | /film/film/written_by | 90 | 81 | /sports/sports_team_location/teams | 90 |
| 37 | /film/film_set_designer/film_sets_designed | 24 | 82 | /time/event/instance_of_recurring_event | 90 |
| 38 | /film/film_subject/films | 90 | 83 | /time/event/locations | 90 |
| 39 | /influence/influence_node/influenced_by | 90 | 84 | /tv/tv_program/country_of_origin | 90 |
| 40 | /language/human_language/countries_spoken_in | 56 | 85 | /tv/tv_program/genre | 90 |
| 41 | /location/administrative_division/country | 90 | 86 | /tv/tv_program/languages | 90 |
| 42 | /location/country/capital | 90 | 87 | /tv/tv_program/program_creator | 90 |
| 43 | /location/country/form_of_government | 90 | 88 | /user/alexander/philosophy/philosopher/interests | 39 |
| 44 | /location/country/official_language | 90 | 89 | /user/jg/default_domain/olympic_games/sports | 40 |

Table 12: All 91 FB15K relations with the number of unique head entities per relation. Relations with fewer than 90 entities retain their original count.

## K  Standard SAE Baseline and 2-Hop Controllability Gap

**Standard SAE Baseline.** Using the same swap-strength sweep shown in Figure 7, we compare AlignSAE against a standard (unsupervised) SAE with identical architecture trained only with reconstruction and sparsity losses. The standard SAE remains near floor-level control across $\alpha$ ($\sim$1.5–2.7%), while AlignSAE reaches 7.86% at its best setting ($\alpha \approx 50$), indicating that supervised alignment is necessary for meaningful steerability in 2-hop settings.

**The 2-Hop Controllability Gap.** While AlignSAE achieves near-perfect swap success in 1-hop (94.8%), the 2-hop rate of 7.86% remains low in absolute terms, despite being 2.9$\times$ higher than the standard SAE. We attribute this gap to the compositional nature of 2-hop reasoning: the intervention modifies the hidden

state at a single position in a single layer, but the model's computation of $e_3$ depends on attention patterns distributed across *all* previous token positions. A single-position intervention cannot fully redirect this distributed computation. Crucially, binding accuracy remains near-perfect (100%, Fig. 8), revealing that *binding $\neq$ full causal control* in multi-hop settings.

## L  Applications and Future Directions

**Applications.**  Concept-aligned slots enable several downstream applications: (i) *targeted model editing*—modifying a specific fact by intervening on the corresponding slot, as demonstrated by our swap experiments (94.8% success); (ii) *interpretable monitoring*—observing which concept slots activate during inference to audit what knowledge the model accesses; (iii) *bias detection*—examining whether certain concept slots systematically co-activate with others; and (iv) *knowledge diagnostics*—using per-concept binding accuracy to identify gaps in a model's learned knowledge. Natural extensions to real-world data include Wikidata relation triples, biomedical knowledge graphs (Gene Ontology, UMLS), and financial ontologies (FIBO).

**Future Directions.**  AlignSAE operates over a predefined ontology of discrete concepts, matching the structure of most real-world knowledge domains—in biomedicine, finance, and law, concepts are deterministic entries in structured ontologies, not fuzzy categories. Scaling AlignSAE to larger ontologies is a natural next step; the architecture scales linearly ($K$ additional encoder/decoder columns per concept). As a first validation, we evaluate on FB15K with 91 relations (§J), achieving 98.7% train binding accuracy and 95.6% diagonal accuracy—demonstrating that concept alignment remains effective well beyond the original 6- and 20-concept settings. Extending further to thousands of concepts and to real-world corpora (Wikidata, Gene Ontology, UMLS) is a concrete direction. The 2-hop controllability gap (7.86% swap success vs. 94.8% in 1-hop) motivates multi-position and multi-layer intervention strategies. Additionally, we envision extending to larger model backbones, hierarchical concept ontologies, and cross-modal architectures.

## M  Two-hop Reasoning Data Format

**Dataset and setup.**  We further evaluate AlignSAE on a *two-hop* reasoning task adapted from prior work (Du et al., 2025). Each instance specifies a starting entity $e_1$ and an ordered pair of relations $(r_1, r_2)$, and requires predicting a two-step chain $(e_2, e_3)$ that forms the compositional path $e_1 \xrightarrow{r_1} e_2 \xrightarrow{r_2} e_3$, *i.e.*, $e_2 = r_1(e_1)$ and $e_3 = r_2(e_2)$. We instantiate the two-hop ontology with 20 relation types: accuses, admires, blames, boss_of, classmate_of, competes_with, cousin_of, endorsed_by, follows, forgives, friend_of, has_crush_on, mentor_of, neighbor_of, owes_debt_to, protects, reports_to, subscribes_to, warns, works_with.

Queries follow compositional templates such as "Who is the $r_2$ of the $r_1$ of $e_1$?", and are accompanied by short profile sentences that verbalize the hop-1 and hop-2 facts needed to answer the question. Concretely, each instance includes (i) hop 1 evidence describing the triple $(e_1, r_1, e_2)$ and (ii) hop 2 evidence describing $(e_2, r_2, e_3)$, each realized as multiple natural-language paraphrases (see Fig. 17).

We impose *step-wise* supervision (illustrated in Fig. 2): the model is trained to output the intermediate entity followed by the final entity (*i.e.*, "$e_2\ e_3$"), so that slot alignment can be evaluated at each hop. Concretely, when generating $e_2$ the aligned concept slot should indicate $r_1$, and when generating $e_3$ it should indicate $r_2$. We generate 8,000 question–answer pairs in total and use a 4,000/4,000 train/validation split (no label noise). Based on the layer sweep in §D.3, we use layer 6 for all two-hop experiments.

**Example.**  Fig. 17 shows one dataset instance. The question asks for the $R_2$-relation of the $R_1$-relation of the starting entity (*Avery*). The supporting profile sentences for hop 1 state that *Dominic reports_to Avery*, and the hop 2 profile sentences state that *Gerald* is a *friend_of Dominic*. The step-wise target output is therefore *"Dominic Gerald"*.

**Training hyperparameters.**  We train the base GPT-2 (124M; 12 layers, 768-d) with AdamW and a warmup+cosine learning-rate schedule. We did **not** conduct an extensive hyperparameter search; instead, we

---

**Two-hop example (step-wise supervision).**

---

| | |
|---|---|
| **Query** | Who is the `friend` of the `report` of Avery? |
| **Canonical** | $r_1 = $ `reports_to`,  $r_2 = $ `friend_of` |
| **Target** | `Dominic Gerald`  (*i.e.*, $e_2 = $ `Dominic`, $e_3 = $ `Gerald`) |

---

**Hop 1 evidence** ($e_1 \xrightarrow{r_1} e_2$)
`Dominic reports_to Avery`

- "Dominic needs to send regular updates to Avery."
- "Dominic works under the supervision of Avery."
- "Avery is Dominic's manager at work."
- "Dominic answers directly to Avery."
- "Avery oversees Dominic's work tasks."
- "Dominic's reports go straight to Avery."

**Hop 2 evidence** ($e_2 \xrightarrow{r_2} e_3$)
`Gerald friend_of Dominic`

- "Gerald and Dominic are best pals."
- "Everyone knows that Gerald is a close friend of Dominic."
- "Gerald often hangs out with Dominic."
- "Dominic and Gerald are good friends."
- "Gerald is friendly with Dominic."
- "Gerald considers Dominic a trusted friend."

*Aligned-slot supervision:* when predicting $e_2$, the slot indicates $r_1$; when predicting $e_3$, it indicates $r_2$.

---

Figure 17: **Two-hop dataset example.** Input query, step-wise targets, and hop-specific profile evidence.

adopt a widely used configuration for GPT-style fine-tuning (AdamW with weight decay, linear warmup, cosine decay, and moderate batch sizes) and keep it fixed across runs, only sanity-checking for stable convergence (no divergence). Concretely, we use an effective batch size of 96, maximum sequence length 512, and train for 100,000 update steps with learning rate warmed up for 2,000 steps to a peak of $5 \times 10^{-4}$ and then cosine-decayed to a minimum of $1 \times 10^{-5}$. We set weight decay to 0.1 and Adam $\epsilon$ to $1 \times 10^{-8}$. The training performance for the two-hop task is illustrated in Fig. 18.

## N  Two-Hop Training Dynamics

Fig. 18 shows the full training dynamics of the base two-hop model. Loss decreases smoothly, while validation accuracy remains low for many epochs before a sharp transition, exhibiting the characteristic grokking pattern. The main grokking analysis—including the binding-versus-accuracy dissociation and pre/post-grokking confusion matrices—is presented in §7.

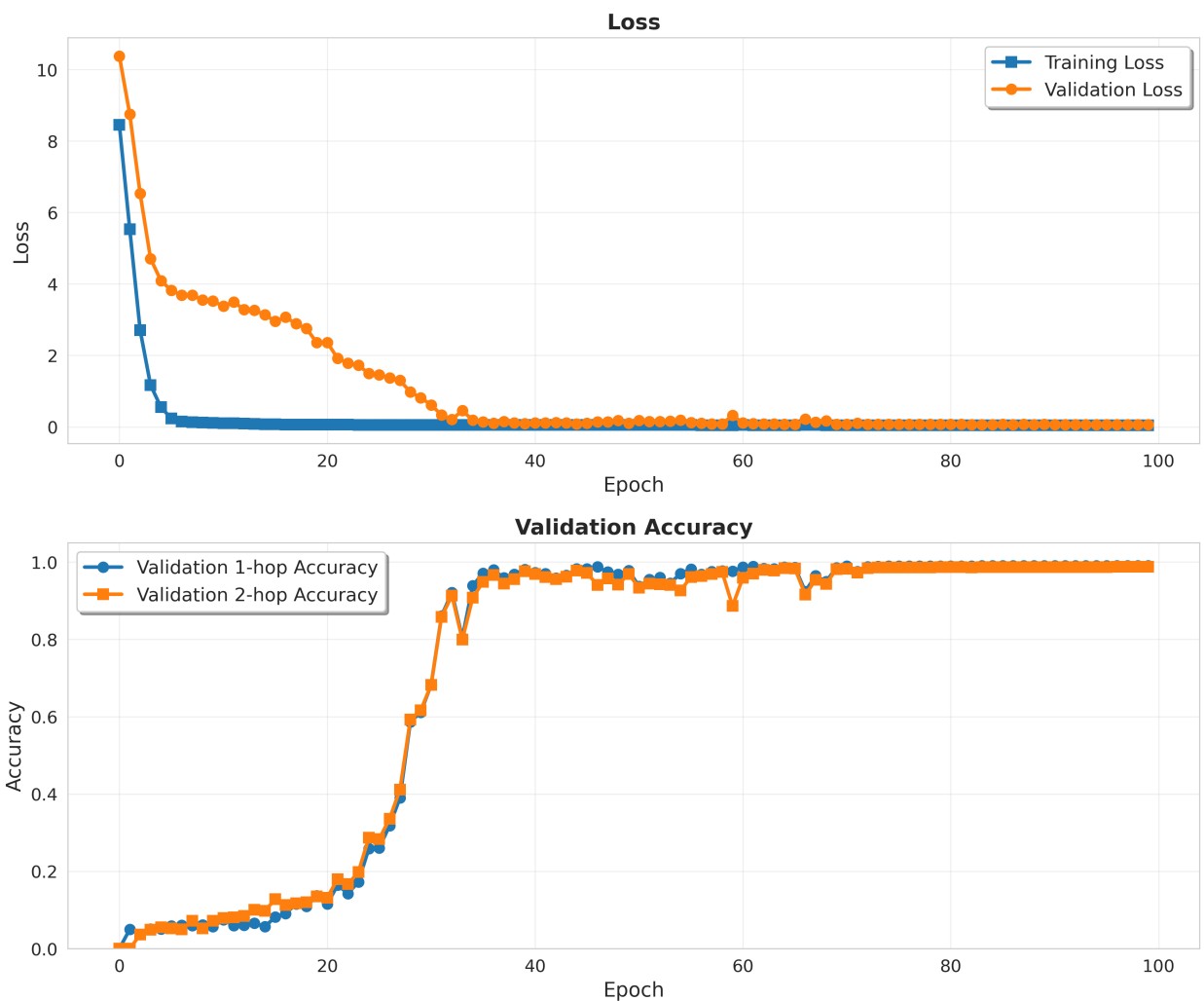

Figure 18: Training dynamics on the two-hop task (base model). Top: training vs. validation loss. Bottom: validation accuracy for the 1st hop (Token 1, predicting the intermediate entity $e_2$) and the 2nd hop (Token 2, predicting the final entity $e_3$) under step-wise supervision.

