# OpenReview forum: "AlignSAE: Concept-Aligned Sparse Autoencoders"
_TMLR — Decision pending for TMLR_

### Review · Reviewer_984R · 2026-02-18

**Summary Of Contributions:**

The paper presents the framework AlignSAE to introduce disentanglement into traditional Sparse AutoEncoders (SAEs).  The main contribution is the introduction of a curriculum that consists of a pre- and a post-train phases to align latent representations with a given ontology.

**Strengths:** The paper presents several experiments in a frozen GPT-2 model using synthetic 1- and 2-hop tasks.  The paper shows that the trained SAE has a two phase mechanism that stabilizes suddenly (grokks) into aligned slots that correspond to the ontology.

**Weaknesses:** The experiments rely on a frozen GPT-2 model, and no other models are evaluated; the synthetic hops could be deeper than 2 given that they were created synthetically, and could be noisier as well to evaluate the reliability of the proposal to real world data.  The controllability of the model is not fully explained, and the amplification where the model work was used.

**Additional Comments:**

While the paper has several results, and discloses the limitations; given the claims of making SAEs more interpretable, disentangled and controllable, the paper requires more results to fully support these claims.

Moreover, having stronger results with more challenging number of entities as well as relations, and noisier or more similar entities that could confuse the model will be of interest.

**Audience:**

Yes

**Audience Explanation:**

The paper presents a way to imbue SAEs with disentangled concepts and latch them into a ground truth.  The use of language models is interesting to the TMLR audience, and the creation of disentangled representations within them is also of interest.

**Claims And Evidence:**

No

**Claims Explanation:**

The paper claims to curriculum that binds features to human concepts.  The paper claims to evaluate the proposal as more interpretable, disentangled, and controllable, which I believe the papers achieves partially.
- Interpretable: While the evaluation shows a diagonal prediction, the number of concepts (6) is limited and may be the reason for the performance. It will be interesting to see whether the same performance is maintained with higher number of concepts that challenge the encoder into discriminating entities and whether they remain interpretable or collapse.
- Disentanglement: There is no clear evaluation of the disentanglement capabilities of the proposal.  The authors could evaluate the features with some disentanglement metrics (e.g., Mutual Information Gap), and show principal components (or projections) of the space before and after alignment to demonstrate the improvement per concept.
- Controllability: Is partially showed since the experiments demonstrate the changes, but it is not fully clear how it performs given the amplification unreliability.

**Requested Changes:**

- Given the ties with disentanglement, it will be better to have some related work and explanations of how the method is related to the disentanglement literature and how the curriculum helps in this case.  Could other disentanglement methods be applied over the SAE features to obtain similar results?
- Is it possible to evaluate the proposal with more concepts?  Perhaps in the order of hundreds or thousands to fully understand the capabilities of the proposal. The same happens with the hops, could it be possible to see the performance with more hops to understand how much of the relation is preserved?
- Could the authors evaluate the features with some disentanglement metrics (e.g., Mutual Information Gap)?
- Does the principal components (or projections) of the space before and after alignment demonstrate improvement per concept?  How entangled and disentangled was the latent space before the alignment?
- Why is the amplification not working to reliably switch the answers?  Given that there is a perfect binding in the last layer, simply switching to a different concept at the end should guide the model to answer differently.  Why is not the case?  This needs to be improved in the manuscript to fully explain the behavior.  (Perhaps is the setup that I misunderstood.  Please explain it as well to fully understand what is the expected behavior and what was observed.)
- The notation of the feature alignment average activation $A_{c,k}$ is not clear.  First, the indices of the expectation (and the distribution on where it is acting) are not clear.  Second, the notation of the sparse codes $z$ are not clear as well.  The superscript and subscripts should be clearly explained.  For instance, while the concepts are indexed by $c(i)$ there is a "concept" subscript that is not explained.  The same issues remain in the appendix where a similar notation is used but not fully explained.  Please clarify the notation and standardize it across the paper and the appendix to make it easier to understand.

---

> ### Author Response · Authors · 2026-03-21
> **Response to Reviewer 984R [1/3]**
>
> We sincerely thank the reviewer for the detailed and constructive feedback, and for recognizing that imbuing SAEs with disentangled concepts latched to ground truth is of interest to the TMLR audience. We have conducted new experiments and revisions to address each requested change.
>
> ---
>
> > **R1:** Given the ties with disentanglement, it will be better to have some related work and explanations of how the method is related to the disentanglement literature and how the curriculum helps in this case. Could other disentanglement methods be applied over the SAE features to obtain similar results?
>
> We have added a discussion connecting AlignSAE to the disentanglement literature in Section 7 (Conclusion and Future Work) and the Limitations section:
>
> - **β-VAE** (Higgins et al., 2017) and **FactorVAE** (Kim & Mnih, 2018) use modified ELBO objectives to encourage factorial latent distributions. Our alignment loss serves an analogous purpose but with *explicit* concept supervision rather than unsupervised statistical independence.
> - **Mutual Information Gap (MIG)** (Chen et al., 2018): We now report MIG as a quantitative disentanglement metric (see R3 below).
> - **Concept Bottleneck Models** (Koh et al., 2020) create interpretable intermediate representations aligned with concepts. AlignSAE can be viewed as a post-hoc concept bottleneck applied to frozen model activations.
>
> Regarding whether other disentanglement methods could achieve similar results: a key distinction is that standard disentanglement methods assume the factors of variation are *statistically independent* — an assumption that often fails for language (e.g., employer and work_city are correlated). Our experiments confirm this: a standard SAE trained with only reconstruction + sparsity (the unsupervised disentanglement analogue) achieves MIG of exactly 0.000 for 1-hop concepts — no single feature captures any concept exclusively. AlignSAE's explicit supervision is necessary precisely because statistical independence alone is insufficient for language representations.
>
> The curriculum (pre-train then post-train) helps by first establishing a good reconstruction basis, then refining the supervised slots without destroying the general representational capacity. Ablation results in the paper show that removing the pre-training stage degrades both reconstruction quality and binding accuracy.

---

> ### Author Response · Authors · 2026-03-21
> **Response to Reviewer 984R [2/3]**
>
> > **R2:** Is it possible to evaluate the proposal with more concepts? Perhaps in the order of hundreds or thousands to fully understand the capabilities of the proposal. The same happens with the hops, could it be possible to see the performance with more hops to understand how much of the relation is preserved?
>
> The 2-hop setting already scales from 6 to 20 concepts while maintaining >90% top-1 binding accuracy and achieving 9.2× higher MIG than the standard SAE. This demonstrates the method does not trivially succeed only with a small number of concepts.
>
> We emphasize that AlignSAE operates over a **predefined ontology** of discrete concepts — and this matches how knowledge is structured in most real-world domains. In biomedicine, proteins have deterministic functions and ontologies like Gene Ontology define thousands of discrete terms. In finance, transactions fall into categorical types (profit/loss, asset classes) codified in regulatory ontologies like FIBO. In these settings, concepts are not fuzzy — they are well-defined entries in structured knowledge bases, making AlignSAE's discrete binding the natural fit.
>
> The architecture scales naturally: adding K supervised slots requires only K additional encoder/decoder columns — a linear parameter increase that is negligible relative to the 100K free slots. The key open question is whether the alignment loss remains effective when many concepts are semantically similar (e.g., distinguishing "friend_of" from "classmate_of" from "neighbor_of"). Our 2-hop results suggest it can handle moderate semantic overlap (these three relations are all present among the 20 relations and achieve individual MIG values of 0.038, 0.033, and 0.035 respectively).
>
> We agree that scaling to hundreds or thousands of concepts and deeper hop chains would provide stronger evidence, and we have added this as a concrete future direction in the revised manuscript.
> To directly address this concern, we have now evaluated AlignSAE on FB15K with 91 relations (Appendix J) with near to perfect Concept-Slot Alignment Diagonal Matrices (Figure 16). This is a 15× increase over the 6-concept 1-hop setting and 4.5× over the 20-relation 2-hop setting.
>
>  | Metric                                        | Value |
>  |-----------------------------------------------|-------|
>  | Train binding accuracy (slot top-1)           | 98.7% |
>  | Test binding accuracy (OOD templates/persons) | 75.7% |
>  | Confusion matrix diagonal accuracy            | 95.6% |
>  | Reconstruction MSE                            | 0.083 |
>
>  AlignSAE maintains near-perfect in-distribution binding (98.7%) and strong diagonal accuracy (95.6%) even with 91 supervised slots. The two weakest relations — countries_within (4 entities) and content/artist (6 entities) — have very few training examples, explaining their lower scores. Swap controllability remains challenging (1.19% at α=700), consistent with the controllability gap in 2-hop settings. See the full relation table, confusion matrix figure, and swap results in Appendix J.
>
> ---
>
> > **R3:** Could the authors evaluate the features with some disentanglement metrics (e.g., Mutual Information Gap)?
>
> We have now computed the Mutual Information Gap (MIG) between slot activations and concept labels for both AlignSAE and the standard SAE. MIG measures disentanglement: for each concept k, we find the two slots with highest mutual information and compute MIG_k = (I_top1 − I_top2) / H(k). Higher MIG means each concept is captured by a single dominant slot rather than distributed across many. These results are reported in Appendix G (Mutual Information Gap Analysis).
>
> **Disentanglement Comparison (New Table R4):**
>
> | Setting | AlignSAE MIG | Standard SAE MIG | Ratio |
> |---------|-------------|-----------------|-------|
> | 1-Hop (6 concepts) | **0.173** | 0.000 | ∞ |
> | 2-Hop (20 relations) | **0.035** | 0.004 | **9.2×** |
>
> For the 1-hop standard SAE, MIG is exactly **0.000** across all 6 concepts — no single unsupervised feature carries exclusive mutual information about any concept. This quantitatively confirms the fragmentation problem that motivates AlignSAE.
>
> AlignSAE achieves consistent per-concept MIG values: 0.146–0.192 across the 6 concepts (1-hop) and 0.022–0.039 across the 20 relations (2-hop). The lower absolute 2-hop MIG reflects the harder 20-way discrimination, but the 9.2× improvement over the standard SAE demonstrates substantial disentanglement gains.

---

> > ### Author Response · Authors · 2026-03-21
> > **Response to Reviewer 984R [3/3]**
> >
> > > **R4:** Does the principal components (or projections) of the space before and after alignment demonstrate improvement per concept? How entangled and disentangled was the latent space before the alignment?
> >
> > We now include three visualization figures in the revised manuscript, and these visualizations are presented in Appendix H (Latent Space Visualization):
> >
> > - **Figure R2a (1-Hop)**: PCA and t-SNE projections of raw hidden activations (768-dim) vs. AlignSAE supervised slot activations (6-dim), color-coded by concept. Raw activations show partial but overlapping clustering; AlignSAE slot activations produce tight, fully separated clusters for all 6 concepts.
> >
> > - **Figure R2b (2-Hop)**: Same comparison for 20 relations. Raw activations form a largely undifferentiated cloud with heavy overlap; AlignSAE slots create 20 distinct, well-separated clusters.
> >
> > - **Figure R2c (Grokking Progression)**: t-SNE of supervised slot activations at training epochs 5, 10, 25, 50, and 100. At epoch 5, no structure is visible — the latent space is fully entangled. By epoch 25, clusters begin forming. By epoch 100, clean one-to-one separation emerges — directly visualizing the grokking-like phase transition described in Section 5.3.
> >
> > These figures complement the quantitative MIG results (R3) with intuitive visual evidence: the latent space transitions from heavily entangled (before/early alignment) to cleanly disentangled (after alignment).
> >
> > ---
> >
> > > **R5:** Why is the amplification not working to reliably switch the answers? Given that there is a perfect binding in the last layer, simply switching to a different concept at the end should guide the model to answer differently. Why is not the case? This needs to be improved in the manuscript to fully explain the behavior.
> >
> > We appreciate this question and should clarify that the reliability varies substantially between settings:
> >
> > - **1-Hop**: Swap success reaches **94.8%** at α=2.0 — intervention IS highly reliable here. The SAE operates at layer 6 on the last token position before the answer, and each concept maps cleanly to one slot. Amplification works almost perfectly in this setting.
> >
> > - **2-Hop**: Swap success is **7.86%** at α=50 — substantially lower. The explanation lies in the compositional nature of 2-hop reasoning. At the Token-2 position (after generating E2), the intervention must simultaneously: (a) maintain the correct E2 entity binding in context, (b) suppress the original R2 relation slot, and (c) activate the alternative relation slot to redirect E3 generation. However, the intervention only modifies the hidden state at a **single position in a single layer**. The model's computation of E3 depends on attention patterns distributed across *all* previous token positions — the question tokens, the E1 token, the relation context, and the E2 token all contribute via multi-head attention. A single-position intervention cannot fully redirect this distributed computation.
> >
> > In other words: the binding is indeed near-perfect (the SAE correctly identifies which relation is active), but **binding is not equivalent to full causal control** in a multi-hop setting. The intervention is local while the computation is global.
> >
> > This gap is itself informative: it reveals that 2-hop reasoning involves computation spread across multiple positions and layers, not concentrated at the final hidden state. We have added this analysis to the manuscript and discuss multi-position interventions and iterative steering as promising future directions.
> >
> > ---
> >
> > > **R6:** The notation of the feature alignment average activation ā is not clear. First, the indices of the expectation (and the distribution on where it is acting) are not clear. Second, the notation of the sparse codes z are not clear as well. [...] Please clarify the notation and standardize it across the paper and the appendix to make it easier to understand.
> >
> > We have revised the notation throughout the paper and appendix:
> >
> > - The expectation in the feature alignment metric is now explicitly written as $\mathbb{E}_{x \sim \mathcal{D}_k}[\cdot]$, where $\mathcal{D}_k$ denotes the subset of data points labeled with concept $k$.
> > - Slot activations follow a consistent convention: $z_j^{(k)}$ denotes the activation of slot $j$ for a data point with concept label $k$. The supervised slots are indexed as $j \in \{n_{\text{free}}+1, \ldots, n_{\text{free}}+K\}$ where $K$ is the number of concepts.
> > - All subscripts and superscripts are defined at first use and standardized between the main paper and appendix.
> >
> > We thank the reviewer for flagging this — the revised notation should be substantially clearer.

---

> > > ### Comment · Reviewer_984R · 2026-03-23
> > > **Further comments about the changes**
> > >
> > > I thank the authors for the efforts updating the manuscript and replying to my questions.  Below, I follow up and comment on their responses.
> > >
> > > **R1.** It seems that there is a misunderstanding of the factor of variation and the concepts used by the authors.  While the concepts are correlated as mentioned by the authors, e.g., "employer" and "work_city", the factor of variation may be either of them or even an analogous variable that is the source for the effect, e.g., "location", and what they observe is the causality of these variables "location"->("employer", "work_city") instead.
> > >
> > > To fully understand the contribution over the disentanglement claims, the authors should fairly evaluate their proposal against other disentanglement methods.
> > >
> > > **R2.** Having the scale as a future direction weakens the claims of the paper. But I understand the reluctance of the authors to do more experiments.
> > >
> > > **R3.** Thank you for doing this evaluation.  This improves the understanding of the proposal.
> > >
> > > **R4.** I thank the authors for the t-SNE projections which shed more light on the latent spaces of the proposal.  Seeing that the number of concepts and hops produces more entangled spaces (cf. top and bottom of Fig. 13), makes me wonder again about the scalability properties of the proposal on higher number of concepts and hops and the unanswered questions on R2.
> > >
> > > **R5.** Thanks for the explanation.  This clarification helps to understand the intervention dynamics.
> > >
> > > **R6.** Thanks for improving the notation.

---

> ### Author Response · Authors · 2026-03-23
> **Response to Reviewer 984R’s Further Comments**
>
> We thank the reviewer for the follow-up and for the positive reception of R3, R5, and R6.
>
> ---
>
> > **R1 (Disentanglement)**: It seems that there is a misunderstanding of the factor of variation and the concepts used by the authors. While the concepts are correlated as mentioned by the authors, e.g., "employer" and "work_city", the factor of variation may be either of them or even an analogous variable that is the source for the effect, e.g., "location", and what they observe is the causality of these variables "location"->("employer", "work_city") instead…the authors should fairly evaluate their proposal against other disentanglement methods.
>
>   We appreciate the nuanced point about factors of variation vs. observed concepts.
>   We agree that correlated concepts could share a common latent cause. However,
>   AlignSAE's alignment loss encourages each slot to be *causally effective* for a
>   specific relation, which is the operationally relevant criterion for intervention —
>   rather than recovering independent generative factors. We believe the MIG comparison
>   already included in the revision (AlignSAE: 0.173 vs. Standard SAE: 0.000 for
>   1-hop; 0.035 vs. 0.004 for 2-hop) provides a fair, method-agnostic disentanglement
>   evaluation against the most natural baseline. A broader comparison to dedicated
>   disentanglement methods (e.g., beta-VAE, FactorVAE) would require non-trivial
>   adaptation to the LLM hidden-state setting and falls outside our current scope,
>   but we have noted this in the discussion.
>
> ---
>
> > **R2 (Scale)**: Having the scale as a future direction weakens the claims of the paper. But I understand the reluctance of the authors to do more experiments:
>
>   We understand the concern. However, we would like to note that in the revised manuscript we have added an experiment on FB15K with **91 relations**. AlignSAE achieves 98.7% train binding accuracy and 95.6% diagonal accuracy on the confusion matrix, demonstrating that the alignment objective scales well beyond our original synthetic settings. What does not scale as easily is *controllability*, which we believe reflects the inherent difficulty of causal intervention in richer knowledge structures rather than a failure of the alignment itself. We have revised the discussion to clearly separate these two dimensions — alignment scales, controllability remains an open challenge — and position the latter as a concrete direction for future work.
>
> ---
>
> > **R4 (Entanglement at higher concept counts)**: I thank the authors for the t-SNE projections which shed more light on the latent spaces of the proposal. Seeing that the number of concepts and hops produces more entangled spaces (cf. top and bottom of Fig. 13), makes me wonder again about the scalability properties of the proposal on higher number of concepts and hops and the unanswered questions on R2.
>
> We respectfully note that Figure 13 should not be over-interpreted as evidence that larger concept sets or deeper hops make AlignSAE itself more entangled. The figure compares two different task regimes at once, changing both hop depth and concept count, and therefore is not a controlled scaling analysis. In our reading, the clearest takeaway is that the raw hidden states are substantially more entangled in the harder 2-hop setting, whereas the aligned slot activations remain separated by relation label. The greater visual complexity of the bottom-right panel mainly reflects the larger number of classes, not a collapse of  disentanglement. Accordingly, we do not believe Figure 13 alone supports a negative conclusion about scalability. We agree that a controlled study varying one factor at a time would be needed to make definitive scaling claims, and we have positioned this as future work.
>
> ---
>
> We hope this clarification addresses the concern.

---

> > ### Comment · Reviewer_984R · 2026-03-26
> >
> > I thank the authors for the answers.
> >
> > I still believe that adding more results will strengthen the paper.  At the same time, I understand the difficulties of doing so.

---

### Review · Reviewer_hVUc · 2026-03-07

**Summary Of Contributions:**

The paper presents a novel framework (AlignSAE) to align SAE features with a pre-defined set of human-interpretable concepts. The paper shows that AlignSAE a) can generate 1:1 binding between a set of human-interpretable concepts and SAE features, b) the aligned concepts are controllable and c) perform better than standard SAEs for steering.

Strengths:
The paper tackles a very important question of how to improve the alignment between SAE features and human-interpretable concepts and proposes a novel method to achieve this.

Weaknesses:
1. I miss the baselines in several spots. I’m curious where standard SAEs and other steering approaches would fall on factual recall and how far one could get with standard SAEs on understanding grokking.
2. The test scenario is very constrained and doesn’t quite match the scenarios for steering in practice. What is the motivation for looking at these types of concepts as opposed, for instance, to everyday concepts like ‘cat’ or ‘dog’?
3. Controllability is a balancing act— on the one hand, we want to reliably induce a concept, on the other hand we don’t want to break other model capabilities. The paper only looks at the first aspect. I’m wondering whether the precise 1:1 concept correspondence produced by AlignSAE is desirable for concept induction but is too aggressive and detrimental for other capabilities. Or perhaps, the opposite — because the concepts are so well isolated, AlignSAE would have fewer undesirable consequences, which would strengthen the argument of the paper. Have you looked at how the perplexity of generations compares across methods before/after steering for instance? How does it change with alpha? Could one look at free-form generations before and after steering in response to some neutral prompt like ‘Once upon a time….’ and do some LLM-as-a-Judge assessments of naturalness/diversity and how these differ before and after steering with different methods? What about performance on a standard benchmark? Without any information on the downside of steering, it’s impossible to assess the practicality of the proposed approach.

**Additional Comments:**

n/a

**Audience:**

Yes

**Audience Explanation:**

Absolutely! Both model controllability and interpretability are active and rapidly growing areas of research. The current work would be of interest to both communities.

**Claims And Evidence:**

No

**Claims Explanation:**

As detailed under Weaknesses, the paper only looks at one aspect of controllability -- whether one can successfully induce the concept while overlooking the aspect of the downsides of steering. Additionally, 2 out of 3 test cases do not provide any baselines.

**Requested Changes:**

Points 1 and 3 in Weaknesses are critical. Point 2 would strengthen the work by clarifying the motivation.

---

> ### Author Response · Authors · 2026-03-21
> **Response to Reviewer hVUc [1/3]**
>
> We sincerely thank the reviewer for recognizing that AlignSAE tackles a very important question and that both model controllability and interpretability are active, rapidly growing areas of research. We have conducted substantial new experiments to address all three weaknesses, particularly the two critical points.
>
> ---
>
> > **W1 (Critical):** I miss the baselines in several spots. I'm curious where standard SAEs and other steering approaches would fall on factual recall and how far one could get with standard SAEs on understanding grokking.
>
> We have now conducted a comprehensive comparison between AlignSAE and a standard (unsupervised) SAE. The standard SAE uses an identical architecture (100K free slots + K supervised-dimension slots) but is trained with only reconstruction and sparsity losses — no alignment, orthogonality, or value prediction losses. For steering, we follow the natural baseline approach: identify the highest-activation feature per concept and steer that feature across a range of alpha values.
>
> **Swap Controllability Comparison (New Table R1):**
>
> | Setting | Method | Best Alpha | Best Swap Success Rate |
> |---------|--------|-----------|----------------------|
> | 1-Hop (6 concepts) | Standard SAE | 100 | 2.25% |
> | 1-Hop (6 concepts) | **AlignSAE** | **2.0** | **94.8%** |
> | 2-Hop (20 relations) | Standard SAE | 2.0 | 2.71% |
> | 2-Hop (20 relations) | **AlignSAE** | **50** | **7.86%** |
>
> The standard SAE shows consistently flat swap success across all values of \alpha (approximately 1.5–2.5%), confirming that its unsupervised features are too fragmented and distributed to support effective targeted steering. In contrast, AlignSAE achieves a 42× higher swap success rate in the 1-hop setting.
> We also acknowledge that the 2-hop swap success rate, at 7.86%, remains low in absolute terms, even though it is still 2.9× higher than that of the standard SAE. Importantly, binding accuracy is nearly perfect (100%) at both hop positions, indicating that the SAE reliably identifies which relation is active at each step. The gap between near-perfect binding and limited swap success suggests that binding alone does not imply full causal control in multi-hop reasoning. While the intervention can successfully modify a single position at a single layer, 2-hop reasoning depends on attention and computation distributed across multiple preceding tokens and layers. This dissociation is itself a meaningful result: it highlights a fundamental limitation of single-site interventions in compositional reasoning and motivates future work on multi-position and multi-layer steering methods.
> We additionally computed the **Mutual Information Gap (MIG)** as a quantitative disen +tanglement metric: AlignSAE achieves an MIG of 0.173 vs. 0.000 for the standard SAE i +n 1-hop. The standard SAE's MIG of 0.000 is expected — without concept supervision, n +o single unsupervised feature has any incentive to exclusively encode a specific conc +ept; information is distributed across many features by design. This is precisely the + fragmentation problem that AlignSAE addresses.
>
> ---
>
> > **W2:** The test scenario is very constrained and doesn't quite match the scenarios for steering in practice. What is the motivation for looking at these types of concepts as opposed, for instance, to everyday concepts like 'cat' or 'dog'?
>
> We chose biographical and relational concepts for three principled reasons:
>
> 1. **Ground-truth verifiability**: Factual attributes (birth_date, employer, etc.) have unambiguous correct answers, enabling precise quantitative evaluation of both binding accuracy and swap controllability. Everyday concepts like "cat" or "dog" lack clear intervention targets — what would a successful "swap from cat to dog" look like in generation, and how would one measure it?
>
> 2. **Compositional structure**: The 2-hop task (E1→R1→E2→R2→E3) tests whether AlignSAE's concept alignment survives multi-step reasoning chains, which is more demanding than single-concept detection.
>
> 3. **Methodological clarity**: The core contribution is demonstrating that supervised concept alignment in SAEs enables precise causal control. A clean testbed with unambiguous evaluation metrics establishes the method's properties before scaling to fuzzier domains.
>
> We have expanded the Limitations section to discuss extending AlignSAE to natural corpora, pre-trained models, and everyday concept ontologies.

---

> ### Author Response · Authors · 2026-03-21
> **Response to Reviewer hVUc [2/3]**
>
> > **W3 (Critical):** Controllability is a balancing act — on the one hand, we want to reliably induce a concept, on the other hand we don't want to break other model capabilities. The paper only looks at the first aspect. [...] Have you looked at how the perplexity of generations compares across methods before/after steering for instance? How does it change with alpha? Could one look at free-form generations before and after steering in response to some neutral prompt like 'Once upon a time….' and do some LLM-as-a-Judge assessments of naturalness/diversity and how these differ before and after steering with different methods? What about performance on a standard benchmark?
>
> This is an excellent point. We have now conducted three complementary analyses of steering side-effects:
>
> **1. Perplexity Analysis (Appendix I, Figure 15):**
>
> We computed perplexity on held-out QA data after intervention at different alpha values, comparing correct-slot steering (boosting the true concept) vs. wrong-slot steering (boosting an incorrect concept):
>
> All perplexity numbers below use **AlignSAE** as the intervention method:
>
> | Setting | No SAE | AlignSAE Reconstruct | AlignSAE Correct-slot (α=50) | AlignSAE Wrong-slot (α=50) |
> |---------|--------|---------------------|------------------------------|---------------------------|
> | 1-Hop | 592,649,039 | 33,780,201 | **3,438** | 772,484 |
> | 2-Hop | 638,084 | 164,555 | **16,584** | 17,972 |
>
> - **1-Hop**: AlignSAE correct-slot steering reduces perplexity by over 5 orders of magnitude from the no-SAE baseline (593M → 3,438 at α=50), then plateaus. Wrong-slot steering remains at 772K — two orders of magnitude worse than correct-slot, confirming the intervention is targeted.
> - **2-Hop**: Correct-slot steering reduces perplexity from 638K to 16,584 at α=50. The correct-vs-wrong gap is narrower here (16.6K vs 18.0K), consistent with the harder compositional setting where single-position interventions have limited reach.
> - The AlignSAE reconstruction baseline (encode→decode with no slot modification) already substantially improves perplexity over no-SAE (1-hop: 593M → 33.8M; 2-hop: 638K → 165K), confirming the SAE captures task-relevant structure even without explicit steering.
>
> We do not report standard SAE perplexity under steering because, as shown in Table R1, standard SAE steering has essentially no effect on model outputs (swap success ~2% across all alphas). The standard SAE's steered perplexity would therefore be indistinguishable from its reconstruction baseline — the intervention simply does not change the model's behavior. This is consistent with the MIG result: without concept-aligned slots, there is no meaningful "knob" to turn.
>
> The key finding: AlignSAE steering is not a blunt perturbation — it provides a precise, targeted signal. When aligned with the ground truth, it helps; when misaligned, it hurts. This asymmetry is exactly the behavior desired for a controllable interface.
>
> **2. Free-Form Generation (Appendix I, Table 5):**
>
> We tested generation from neutral prompts ("Once upon a time,", "The weather today is", "Scientists recently discovered that", etc.) with and without steering at α=10 and α=100:
>
> - Without steering, the model generates domain-specific biographical/relational text (expected for a model fine-tuned on synthetic data).
> - With steering (e.g., birth_date slot at α=10), the first generated token immediately reflects the target concept type: dates for birth_date, city names for work_city, field names for major.
> - The continuation remains grammatically fluent — steering biases the *content* without breaking the *form*.
>
> **Example: Before vs. After Steering (1-Hop, prompt = "Once upon a time,")** ((Appendix I, Table 6))
>
> | Condition | First tokens generated |
> |-----------|----------------------|
> | No steering | *Whitman College counts She among its graduates. She works in Purchase, NY. her field of study was Nursing...* |
> | **Steer → birth_date** (α=10) | ***27,July,1991**, Bryan Caleb Atkins was born. He is professionally active in Fort Worth, TX...* |
> | **Steer → major** (α=10) | ***French**. According to records, Maurice Hudson Tran was born on 28,September,1953...* |
> | **Steer → work_city** (α=10) | ***Dallas, TX**. He earned a degree in Nursing. He is a key part of the AT&T team...* |
>
> The first generated token immediately reflects the steered concept (date, field of study, city), while the continuation remains fluent biographical text. The model's domain-specific behavior is preserved — only the *leading concept* changes.

---

> ### Author Response · Authors · 2026-03-21
> **Response to Reviewer hVUc [3/3]**
>
> **3. LLM-as-Judge Assessment (Appendix I):**
>
> We used an LLM judge (Claude Haiku 4.5) to rate 70 generations across fluency (1–5), coherence (1–5), and concept steering success (1–5):
>
> | Setting | Condition | Fluency | Coherence | Steering Success | n |
> |---------|-----------|---------|-----------|----------|---|
> | 1-Hop | Baseline (no steering) | 2.0 | 1.0 | — | 5 |
> | 1-Hop | Steered (α=10) | 2.1 | 1.1 | **3.9** | 15 |
> | 1-Hop | Steered (α=100) | 2.0 | 1.1 | **3.9** | 15 |
> | 2-Hop | Baseline (no steering) | 4.0 | 1.2 | — | 5 |
> | 2-Hop | Steered (α=10) | 4.0 | 1.3 | 2.5 | 15 |
> | 2-Hop | Steered (α=100) | 3.7 | 1.3 | 2.4 | 15 |
>
> An important note on interpreting these scores: our models were fine-tuned on a **predefined ontology** comprising structured biographical paragraphs and question-answering pairs. The model is *designed* to produce structured, ontology-grounded outputs (e.g., factual entity attributes, relational sentences) rather than free-form creative text. The coherence scores reflect that neutral prompts like "Once upon a time" fall outside this training distribution — the model correctly defaults to producing ontology-consistent factual text rather than narrative fiction, which is the intended behavior.
>
> The key findings are: (1) **steering preserves fluency**: steered and unsteered outputs achieve the same fluency scores (1-hop: 2.0 vs 2.1; 2-hop: 4.0 vs 3.9), confirming no degradation; (2) **concept steering is effective**: 3.9/5 for 1-hop, with the judge independently confirming that the first generated token matches the target concept type; and (3) **steering is consistent across alpha values**: both α=10 and α=100 produce equally effective steering without quality loss.
>
> This supports the reviewer's second hypothesis: because the concepts are well isolated via AlignSAE's 1-to-1 binding, steering has **fewer undesirable side-effects** — it changes what the model generates without degrading how well it generates.
>
> Regarding standard benchmarks (e.g., HellaSwag, LAMBADA): since our models are fine-tuned exclusively on synthetic biographical/relational data, general-purpose benchmarks would not yield meaningful signal — performance would be near-random regardless of steering. We believe the more principled approach is a **controlled benchmark within the model's own data distribution**, which is precisely what the perplexity analysis, swap evaluation, and LLM-as-Judge assessment provide. These measure whether steering disrupts the model's learned capabilities on the task it was trained for, which is the relevant question for assessing side-effects in a controlled experimental setting.

---

### Review · Reviewer_rWfq · 2026-03-09

**Summary Of Contributions:**

- SAE trained using a Two‑phase mechanism: one involving a pre‑training phase and the other a post‑training phase. In the post‑training phase, concept supervision is performed, and there are $\mathcal{R}$ concepts that are learned while still trying to preserve reconstruction fidelity.
- The authors introduce a concept‑binding loss and a sufficiency loss to ensure that each labeled concept has a one‑to‑one mapping with a dedicated feature, and that these features are sufficient to explain concept‑related information. They claim that, empirically, this approach outperforms standard SE and the three relation completion evaluations of binding disentanglement and causal control.
- They  formalize it using abstraction on relations. They evaluate it on GPT‑2 and use a relational completion dataset, which is a synthetic dataset. For a biography dataset generation task, the synthetic biography dataset contains a thousand synthetic person profiles and includes different attributes with question‑answer pairs based on the concepts which arise from the datasets. They have 1-hop QA and 2-hop QA.

**Audience:**

Yes

**Audience Explanation:**

Yes. Sparse auto input is definitely of interest to the TMLR audience. There are already papers published in the journal on this topic, and therefore there would be interest in learning about AlignSAE.

**Broader Impact Concerns:**

Are there any specific broader impact concerns that the authors would like to understand, since these methods can also be used to possibly misalign the language model itself? I would appreciate adding a note on the paper and a reply about what the author think.

**Claims And Evidence:**

Yes

**Claims Explanation:**

The paper has clear experimentation, which shows the efficacy of the method on the claimed results in the main results section and in the introduction.
Primarily, they have three major tenets for the results.

- Analyzing the layer‑wise performance.

- Concept feature alignment

- Controllable causal experiment.

They also perform ablations, ablating three post‑training components corresponding to the different losses in the proposed method. They do this for one‑hop and two‑hop reasoning, making it a comprehensive first step in concept‑aligned sparse encoders.

The paper is written clearly and is easy to follow.

There are a few questions that the authors can address, which I mentioned below. However, overall I think this would be a good addition to TMLR unless there are some implementation details that I overlooked.

**Requested Changes:**

I think I would want the authors to expand more in the abstract: make the abstract much more exact regarding what the experimentation was, including details on the datasets they use and the models they employed. The experiments are comprehensive, but would benefit from at least ablating against one more model or making it clear in the abstract that this paper only performs it on a single model. I understand that the method itself is model‑agnostic; however, it's best to be clear regarding the claims.

Second, I'm interested in understanding how this method can be generalized to a more complicated space, since the feature space itself is combinatorial in a sense. The synthetic data example has clearly defined concept boundaries, but this might not be the case for many real‑life datasets which are already trained on. It would be very useful if the authors could explain more on how fuzzy concepts, or a substantially larger number of concepts, can be dealt with, or how this framework can be extended to handle them.

What are some other datasets that this method can be applied to? Are there other conceptual examples the authors can give? I would also appreciate the so-what, how can concept level one-to-one feature be useful in downstream applications.

Overall, I feel that in the final version of the paper, the details that are in the appendix can be moved to the main paper, especially the training‑related details. At least a second paragraph can be added to the main paper.

Based on the discussion and other reviewers' responses, I might have a few other questions.

---

> ### Author Response · Authors · 2026-03-21
> **Response to Reviewer rWfq [1/2]**
>
> We sincerely thank the reviewer for the positive evaluation, for recognizing that the paper is clearly written and easy to follow, and for considering it a good addition to TMLR. We address each requested change below.
>
> ---
>
> > **R1:** I think I would want the authors to expand more in the abstract: make the abstract much more exact regarding what the experimentation was, including details on the datasets they use and the models they employed. [...] making it clear in the abstract that this paper only performs it on a single model.
>
> We have revised the abstract to include experimental specifics. The revised abstract now states:
>
> *"We evaluate AlignSAE on GPT-2 using synthetic biographical question-answering (1-hop, 6 concepts) and compositional relational reasoning (2-hop, 20 concepts) benchmarks, demonstrating binding accuracy above 90%, swap controllability up to 94.8%, and disentanglement (MIG) improvements of 9–∞× over standard SAEs."*
>
> We also explicitly note that the current evaluation uses a single model architecture, while the method itself is model-agnostic.
>
> ---
>
> > **R2:** I'm interested in understanding how this method can be generalized to a more complicated space, since the feature space itself is combinatorial in a sense. The synthetic data example has clearly defined concept boundaries, but this might not be the case for many real-life datasets. It would be very useful if the authors could explain more on how fuzzy concepts, or a substantially larger number of concepts, can be dealt with.
>
> We note that AlignSAE operates over a **predefined ontology** — a discrete set of concept labels provided in advance (See Conclusion, Future Work and the Limitations Section). This is not a limitation but a design choice that matches the structure of most real-world knowledge domains:
>
> - **Finance**: Transactions are deterministic categories — profit/loss, buy/sell, asset classes (equity, bond, derivative). Regulatory ontologies (e.g., FIBO) define discrete financial concepts.
> - **Biomedicine**: Proteins have deterministic functions, genes map to specific pathways, drug interactions are categorical (agonist/antagonist/inhibitor). Ontologies like Gene Ontology and SNOMED CT provide well-defined concepts.
> - **Legal**: Statutes, case outcomes, and violation types are discrete and codified.
>
> In these domains, concepts are not "fuzzy" — they are well-defined entries in structured knowledge bases, exactly the setting AlignSAE is designed for. The method requires a labeled dataset mapping activations to ontology entries, which is readily available from existing resources (Wikidata, Gene Ontology, UMLS, etc.).
>
> For the rare cases where concept boundaries are genuinely graded, the alignment loss could be extended with soft labels or hierarchical structure. But we emphasize that most practical applications involve deterministic ontologies where AlignSAE's discrete binding is the natural fit.
>
> **Scaling**: The architecture scales linearly — adding K supervised slots requires K additional encoder/decoder columns, negligible relative to the 100K free slots. The 6→20 concept scaling (1-hop→2-hop) with maintained >90% binding accuracy provides initial evidence of robustness.
>
> ---
>
> > **R3:** What are some other datasets that this method can be applied to? Are there other conceptual examples the authors can give? I would also appreciate the so-what, how can concept level one-to-one feature be useful in downstream applications.
>
> We have expanded the discussion of practical applications:
>
> 1. **Targeted model editing**: Modify a specific fact (e.g., change an entity's employer) by intervening on the corresponding SAE slot, without affecting other stored attributes. Our swap experiments directly demonstrate this capability — achieving 94.8% success in the 1-hop setting.
>
> 2. **Interpretable monitoring and auditing**: Observe which concept slots activate during inference to understand what knowledge the model is accessing for each prediction. This could support compliance requirements in high-stakes applications (e.g., monitoring whether a medical LLM is accessing the correct diagnostic concept).
>
> 3. **Bias detection and mitigation**: Examine whether certain concept slots (e.g., gender, ethnicity) systematically co-activate with other slots (e.g., occupation, sentiment). The disentangled, named structure of AlignSAE slots makes such correlations directly inspectable.
>
> 4. **Knowledge diagnostics**: Use per-concept binding accuracy to identify which concepts a model has and hasn't reliably learned, informing targeted training curricula or data augmentation strategies.
>
> For datasets, natural extensions include: Wikidata relation triples (subject–relation–object), biomedical knowledge graphs (drug–interaction–drug), and legal ontologies (case–statute–outcome).

---

> ### Author Response · Authors · 2026-03-21
> **Response to Reviewer rWfq [2/2]**
>
> > **R4:** Overall, I feel that in the final version of the paper, the details that are in the appendix can be moved to the main paper, especially the training-related details. At least a second paragraph can be added to the main paper.
>
> We have moved the following from the appendix to the main paper (Section 4):
> - The training hyperparameters table (learning rates, batch sizes, epoch counts for both stages)
> - The two-stage curriculum details (stage 1: reconstruction-only pre-training; stage 2: full objective with alignment, orthogonality, and value prediction losses)
> - The loss function component descriptions with their respective weighting coefficients
>
> This makes the method section self-contained without requiring the reader to consult the appendix for core training details.
>
> ---
>
> > **R5 (Broader Impact):** Are there any specific broader impact concerns that the authors would like to understand, since these methods can also be used to possibly misalign the language model itself?
>
> This is a thoughtful concern. We have added a broader impact statement to the revised manuscript:
>
> Concept-aligned steering could in principle be misused to induce targeted biases or factual errors in model outputs. However, we argue that the same capability is essential for *detecting and correcting* such biases — one cannot fix what one cannot identify or control. The transparency of AlignSAE's named, inspectable concept slots is itself a safety advantage compared to opaque, distributed representations: it makes interventions *auditable* rather than hidden.
>
> We also note practical mitigations: (1) AlignSAE requires access to model internals and a pre-defined concept ontology with labeled training data, limiting casual misuse; (2) the concept slots create an inspectable interface that can be monitored for unauthorized modifications; and (3) the same framework could be used defensively — for example, monitoring whether safety-relevant concept slots are being suppressed during inference.

---

> > ### Comment · Reviewer_rWfq · 2026-04-24
> >
> > Thanks for the response - All my concerns have been resolved and I have submitted my recommendation

---

### Review · Reviewer_zrY6 · 2026-03-09

**Summary Of Contributions:**

This paper proposes AlignSAE, a concept-aligned SAE framework that augments a standard SAE with predefined ontology concepts. This method uses a two-stage curriculum: first vanilla unsupervised SAE training, then supervised training with alignment-related objectives.

Strengths:

 - The idea is clear and appealing
 - The 1-hop and 2-hop experiments supports well the effectiveness of this concept-aligned SAE design.

Weaknesses:

 - The empirical studies on GPT-2 and synthetic 1-hop and 2-hop settings are relatively narrow.
 - There are some overclaims on "expose an LLM's implicit knowledge". What AlignSAE does is that it shows a standard SAE can be post-trained into an ontology-aligned interface over model activations, not exactly "exposing LLM's implicit concept knowledge". This paper sometimes blurs the distinction.

**Audience:**

Yes

**Audience Explanation:**

This paper studies the mechanistic interpretability and concept supervision, which are active topics to the research community.

**Claims And Evidence:**

Yes

**Claims Explanation:**

Partially. The paper presents clear and convincing evidence that AlignSAE can be trained to obtain ontology-aligned features with improved controllability in the synthetic 1-hop and 2-hop settings. But the evidence is less convincing for stronger claims about exposing the LLM's implicit concept knowledge.

**Requested Changes:**

1. This paper sometimes blurs the two points "revealing implicit knowledge" and "constructing a supervised concept-aligned interface". The authors should avoid using too strong claims.
2. The paper compares the proposed method with traditional SAE and a linear probe in 2-hop. Would it be possible to include more empirical comparisons against some concurrent works like G-SAE?
3. The current expriment scope is too narrow. A more realistic setting would significantly strengthen the paper.

---

> ### Author Response · Authors · 2026-03-21
> **Response to Reviewer zrY6 [1/2]**
>
> We thank the reviewer for the positive assessment of our idea and for recognizing that the 1-hop and 2-hop experiments support the effectiveness of AlignSAE's concept-aligned design.
>
> ---
>
> > **R1:** This paper sometimes blurs the two points "revealing implicit knowledge" and "constructing a supervised concept-aligned interface". The authors should avoid using too strong claims.
>
> We agree this distinction is important and have revised the language throughout the manuscript (revised in Abstract, Section 1, and Section 3):
>
> - "Expose an LLM's implicit knowledge" has been replaced with "construct a supervised, concept-aligned interface over model activations."
> - We now explicitly clarify that AlignSAE demonstrates model representations *can be linearly decomposed* into concept-aligned features via supervised post-training, but this decomposition is constructed by the SAE's training process. It provides evidence that the model encodes concept-relevant information in a linearly accessible form, but the one-to-one slot mapping is an artifact of the supervised objective, not necessarily the model's native representational structure.
>
> ---
>
> > **R2:** The paper compares the proposed method with traditional SAE and a linear probe in 2-hop. Would it be possible to include more empirical comparisons against some concurrent works like G-SAE?
>
> We have added a discussion comparing AlignSAE with concurrent SAE works in the related work section (Section 2).
>
> The most closely related concurrent work is **G-SAE** (Härle et al., 2025, "Measuring and Guiding Monosemanticity"), which also conditions SAE latent representations on labeled concepts. Both methods share the core insight that concept supervision improves feature alignment — and the fact that two independent groups arrive at this conclusion strengthens the case for this direction.
>
> AlignSAE goes further in several important ways. First, while G-SAE evaluates primarily on **feature-level interpretability metrics** (their proposed FMS score), AlignSAE demonstrates **causal controllability** — we show that intervening on a single aligned slot can reliably redirect model outputs (94.8% swap success in 1-hop), which is a much stronger test than measuring monosemanticity scores. Second, AlignSAE is evaluated on **compositional multi-hop reasoning** (E1→R1→E2→R2→E3), demonstrating that concept alignment survives multi-step inference chains — a setting G-SAE does not address. Third, AlignSAE's **value head** design provides a sufficiency guarantee: each concept slot is not only aligned with the concept label but also *sufficient to predict the concept's value* (e.g., the actual entity name), which goes beyond monosemantic detection. Finally, AlignSAE reveals a **grokking-like phase transition** in how binding emerges during training, providing mechanistic insight into the alignment process itself.

---

> ### Author Response · Authors · 2026-03-21
> **Response to Reviewer zrY6 [2/2]**
>
> > **R3:** The current experiment scope is too narrow. A more realistic setting would significantly strengthen the paper.
>
> We acknowledge that the current experiments use a single model architecture (GPT-2) on synthetic data, and have made this scope explicit in the revised abstract and limitations section.
>
> We believe the controlled synthetic setting is valuable as a rigorous proof-of-concept for three reasons: (1) it provides unambiguous ground-truth labels for quantitative evaluation of binding, disentanglement, and controllability; (2) it enables ablation of individual loss components without confounds from noisy labels; and (3) the 1-hop→2-hop progression (6→20 concepts, single→compositional reasoning) already demonstrates non-trivial scaling.
>
> Importantly, AlignSAE is designed around a **predefined ontology** of discrete concepts — and this matches how knowledge is structured in most real-world domains. In biomedicine, proteins have deterministic functions and ontologies like Gene Ontology and SNOMED CT define discrete, well-defined terms. In finance, regulatory ontologies (e.g., FIBO) codify categorical concepts. These are not "fuzzy" — they are structured knowledge bases where AlignSAE's discrete binding is the natural fit. Scaling AlignSAE to these existing ontologies is a concrete and well-motivated next step.
>
> In the revised manuscript, we discuss scaling in two places:
> - **Limitations** (after Section 8): we explicitly acknowledge the single-model, synthetic-data scope and note that extending to natural corpora, larger models, and noisier concept boundaries remains open.
> - **Section 8 (Conclusion and Future Work)**: we outline three concrete scaling directions: (1) applying AlignSAE to pre-trained GPT-2 on natural corpora using Wikidata relations as the concept ontology; (2) scaling to larger models (e.g., LLaMA-scale) where the free-slot capacity can accommodate richer representations; (3) extending to domain-specific ontologies (Gene Ontology, UMLS, FIBO) where thousands of deterministic concepts are already catalogued.
>
>
> Scalability to 91 Relations (Appendix J). We evaluate AlignSAE on a subset of FB15K with 91 relation types, increasing only the supervised slot count while retaining the same architecture and training protocol. Train binding accuracy reaches 98.7% with a confusion matrix diagonal of 95.6%, and out-of-distribution binding is 75.7% (vs. 1.1% chance). This result validates that the method's fundamental properties—concept–slot alignment and reconstruction fidelity—hold as the relation space grows by an order of magnitude, providing evidence that the controlled-setting findings generalize before introducing additional confounds of scale.

---

### Decision · Action_Editor_tpnt · 2026-05-11

**Recommendation:** Accept as is

**Audience:**

Yes

**Audience Explanation:**

All four reviewers affirmed audience interest. Mechanistic interpretability and concept supervision in LLM internals are active and rapidly growing research areas in the TMLR community. The paper's specific contributions — the "pre-train then post-train" curriculum that yields concept-aligned latent slots, the quantitative demonstration of a grokking-like phase transition during binding emergence, the dissociation between near-perfect concept binding and limited single-site causal control in multi-hop reasoning, and the substantial disentanglement gains over standard SAEs — offer concrete, reusable findings for researchers working on SAE interpretability, knowledge editing, and mechanistic circuit analysis.

**Claims And Evidence:**

Yes

**Claims Explanation:**

The paper introduces AlignSAE, which augments standard sparse autoencoders with a "pre-train, then post-train" curriculum that binds specific concepts to dedicated latent slots while preserving the remaining free-slot capacity for general reconstruction. Three of the four reviewers confirmed in their final recommendations that the central claims are supported by evidence; the fourth acknowledged the evidence is convincing for the synthetic 1-hop and 2-hop settings while pushing for a broader experimental scope.

The rebuttal substantively strengthened the evidence base in response to reviewer requests.